# A *Bacteroides thetaiotaomicron* genetic locus encodes activities consistent with mucin *O*-glycoprotein processing and *N*-acetylgalactosamine metabolism

Didier A. Ndeh [1] ✉, Sirintra Nakjang[2], Kurt J. Kwiatkowski[3], Claire Sawyers[4], Nicole M. Koropatkin [3], Robert P. Hirt [4] ✉ & David N. Bolam [4] ✉

The gut microbiota is a key modulator of human health and the status of major diseases including cancer, diabetes and inflammatory bowel disease. Central to microbiota survival is the ability to metabolise complex dietary and host-derived glycans, including intestinal mucins. The prominent human gut microbe *Bacteroides thetaiotaomicron (B. theta)* is a versatile and highly efficient complex glycan degrader thanks to the expansion of gene clusters termed polysaccharide utilisation loci (PULs). While the mechanism of action for several singular dietary glycan-induced PULs have been elucidated, studies on the unusually high number of mucin-inducible PULs in *B. theta* significantly lag behind. Here we show that a mucin inducible PUL BT4240-50 encodes activities consistent with the processing and metabolism of mucin *O*-glyco-proteins and their core sugar *N*-acetylgalactosamine (GalNAc). PUL BT4240-50 was also shown to be important for competitive growth on mucins in vitro, encoding a kinase (BT4240) critical for GalNAc metabolism. Additionally, BT4240-kinase was shown to be essential for glycosaminoglycan metabolism, extending the PULs function beyond mucins. These data advance our understanding of glycoprotein metabolism at mucosal surfaces, highlighting GalNAc as a key metabolite for competitive microbial survival in the human gut.

The majority of the human gut microbiota resides in the colon, with the colonic microbiota increasingly recognised as a metabolic organ that profoundly influences numerous aspects of human biology[1]. Critical to microbiota-colon homoeostasis is the mucus layer that provides protection for the underlying epithelia from the huge microbial load[2–4]. Colonic mucus is composed of two discrete layers, the looser outer layer is a niche for some members of the microbiota and is associated with the essentially microbe-free and more compact and rigid inner layer[2–5] (Fig.1A). These two strata are essential in maintaining digestive and immunological homoeostasis where human enterocytes and immunocytes coordinate their activities with resident microbes for mutually beneficial outcomes[4,6,7].

Complex glycans are the major nutrient source available to the colonic microbiota from dietary and host-derived sources. Dietary sources include ingested plant, microbial and animal material, while host glycans are mainly mucosa associated macromolecules including mucins and glycosaminoglycans[7–9]. Mucins are heavily *O*-glycosylated proteins that represent the major organic components of

[1]Division of Plant Sciences, School of Life Sciences, University of Dundee, Dundee, UK. [2]Precision Medicine Centre of Excellence, Queen's University Belfast, Belfast, UK. [3]Department of Microbiology and Immunology, University of Michigan Medical School, Ann Arbor, MI, USA. [4]Biosciences Institute, Medical School, Newcastle University, Newcastle upon Tyne NE2 4HH, UK. ✉e-mail: dndeh001@dundee.ac.uk; robert.hirt@newcastle.ac.uk; david.bolam@newcastle.ac.uk

**Fig. 1 | Mucins and various mucin cores structures showing the context and location of the core and terminal sugar GalNAc. A** Mucins are the major organic components of mucus on mucosal surfaces and consist of a peptide backbone consisting of multiple PTS repeats and heavily decorated by *O*-linked glycans resulting in a 'bottle brush'-like appearance of mucin chains[2,10,11,40]. GalNAc is the hallmark and core of mucin *O*-glycans, forming the first *O*-glycosidic linkage to the peptide with the side chain of serine or threonine (only Ser shown here) as shown in the model mucin glycan structure. GalNAc is also present as a terminal decoration of mucin side chains such as blood group A antigen (BgA) (shown top left of an example mucin glycan chain). Colonic mucus has a two layer structure; the looser outer layer is a niche for the microbiota and is associated with the essentially microbe-free and more compact and rigid inner layer[2]. **B** Mucin *O*-glycan side-chains are highly variable, but are initially composed of eight known core structures[12], with cores 1-4 being the most common in intestinal mucins. Core 3 is common in colonic mucin (mainly MUC2), whereas core 1 is more common in the secreted mucins of the upper digestive tract (e.g. MUC5AC, MUC6)[11,40,57].

gut epithelium glycocalyx and mucus, comprising of a central hydroxy-amino-acid rich polypeptide backbone (often referred to as the PTS repeat region for proline-threonine-serine) with *O*-linked glycan side chains and in gel-forming mucins, N- and C-terminal cysteine rich domains that cap each end and facilitate cross-linking of the chains to form a complex mucin network[2,10] (Fig. 1A). Central to all *O*-glycoproteins is the core sugar GalNAc which links the glycan chain through an *O*-glycosidic bond to serine or threonine in the PTS backbone. In mucins, the resulting *O*-glycan side chains are characterised by considerable structural diversity of GalNAc-containing core structures, backbone repeats and terminal epitopes, resulting in a highly complex and heterogeneous macromolecule[10–12] (Fig. 1A, B). Notably, only a subset of bacterial species of the gut have developed the capacity to graze on mucins, a trait thought to represent a high level of adaptation to this niche, facilitating both initial colonisation and long-term survival during absence of diet-derived glycans and thereby considered to play a key role in community development and stability[7,9,11,13–18].

In order to maintain digestive and immunological homoeostasis, mucin production and grazing must be balanced, with the microbiota contributing to both of these processes[10,11]. Significantly, it is increasingly recognised that the outcome of unbalanced mucin grazing by the microbiota plays a key role in disease both locally, as in inflammatory bowel diseases (IBD) and colon cancer, and more systemically, such as in diabetes and arthritis[1,4,9–11,19–21]. As a result, there has been considerable interest in recent years to improve our understanding of the mechanisms of mucin *O*-glycoprotein metabolism by members of the gut microbiota. The prominent human colonic microbe *B. theta* is characterised by extensive glycan degradation capabilities, a broad generalist capable of targeting plant, microbial and animal-derived substrates, including mucins[8,17,22]. Indeed, the ability of *B. theta* to access mucins has been shown to be a key trait required for effective colonisation and maintenance of this microbe in a mouse model[17]. In Bacteroidota, the major gram-negative phylum of the gut, the glycan degrading apparatus is encoded by discrete polysaccharide utilisation loci (PULs); co-regulated gene clusters that each encode a suite of

cell-envelope located carbohydrate-active enzymes (CAZymes)[23] and glycan-binding and transport proteins that orchestrate the acquisition, import and degradation of a specific complex glycan[8,24–27]. In *B. theta* 16 to 18 PULs encompassing ~100 genes are up-regulated on exposure to mucin *O*-glycans, either in vitro or in vivo; currently representing the largest number of PULs dedicated to a specific class of glycan[17,28,29]. The large number of PULs involved not only emphasises the importance of mucins utilisation to *B. theta* survival, but also the complexity and heterogeneity of these substrates. In addition, only a very small number of the induced genes have been functionally characterised[21,30,31] highlighting our limited understanding of mucin metabolism at mucosal surfaces.

We originally identified M60-like (M60L) peptidases as a novel subfamily of metallopeptidases significantly enriched among mucosal microbes, including *B. theta* and demonstrated that the *B. theta*-derived M60-like domain-containing protein BT4244 (BT4244-M60L) possesses metal dependent mucolytic activity in vitro[32]. Subsequent studies investigated the structural and molecular basis of this activity, establishing that the enzyme specifically functions as an *O*-glycopeptidase with potential applications in mucin detection/visualisation studies[33–37]. BT4244-M60L is part of a PUL termed PUL BT4240-50 that is up-regulated during growth of *B. theta* on mucin both in vitro and in vivo and one of five *B. theta* PULs that is important in facilitating colonisation of the mouse gut[17,29,38,39]. The BT4240-50 PUL contains several other genes whose function are unknown. In a bid to enhance our understanding of the functional context of BT4244-M60L and ultimately mucin metabolism in *B. theta*, we characterised PUL BT4240-50 using a combination of genetic, biochemical, structural and cell biology approaches. We show that PUL BT4240-50 encodes a machinery that couples the processing of mucins with the metabolism of GalNAc, a core and abundant mucin *O*-glycan sugar. Within this machinery, various components are seen to exhibit complementary binding and enzymatic activities consistent with the processing of core 1 (Galβ1,3GalNAc) containing glycoproteins (Fig. 1). Genetic studies further reveal an essential role for the PUL in the competitive metabolism of mucins in vitro. The data also shows that PUL BT4240-50 encodes a critical GalNAc kinase enzyme which extends the PULs metabolic function beyond mucins by virtue of its ability to phosphorylate GalNAc from diverse sources. Overall, these data support a model where GalNAc acquisition and metabolism mediate competitive growth of gut *Bacteroides* species in the presence of mucin and other GalNAc-containing macromolecules, revealing further avenues to dissect the complex mucosa-microbiota interactions in the gut and their impact on health and disease.

## Results

### PUL BT4240-50 encodes proteins predicted to be involved in mucin breakdown

The PUL BT4240-50 (Fig. 2A) is upregulated on exposure of *B. theta* to mucins both in vitro and in vivo, suggesting the encoded proteins are involved in the acquisition and breakdown of a discrete structure in these complex glycoproteins[17,38,39]. Notably, the PUL has also been shown to be specifically upregulated during growth on core 1 disaccharide (Galβ1,3GalNAc), a core component of various mucins[17,40] (Fig. 1A, B). The PUL is divided into two major operons BT4240-43 and BT4244-47 (Fig. 2A). The former operon (operon 1) is constitutively transcribed at a high basal level when cells are grown on glucose and up-regulated ~11-fold (average across ORFs) in the presence of porcine gastric mucin III (PGMIII) whereas operon BT4244-47 (operon 2) has a lower basal level of expression and is up-regulated ~130-fold (average across ORFs) in the same growth conditions[17]. BT4248-50 encode a predicted ECF-type anti-sigma and sigma factor and are likely involved in regulation of the PUL, although the genes themselves are not upregulated during growth on mucin (Fig. 2A). Based on in silico analyses using Pfam, KEGG and CAZy annotations[23,41,42], operon 1 of the

PUL encodes a protein of unknown function (BT4240) with a phosphotransferase enzyme family domain, two glycoside hydrolases (GH) (BT4241 and BT4243) belonging to family GH2 and GH109, respectively and a putative inner membrane sugar transporter (BT4242). Operon 2 encodes BT4244-M60L, a predicted surface lipoprotein previously shown to possess *O*-glycopeptidase activity (Fig. 2A)[32–37], as well as homologues of the outer membrane canonical Sus proteins SusD (BT4246) and SusC (BT4247) known to be involved in cell surface glycan import[25,26,43,44] (Fig. 2A). Downstream of BT4246 is the gene BT4245, predicted to encode a surface glycan binding protein (SGBP) by virtue of the presence of an N-terminal DUF1735 'linker' domain, a family 32 carbohydrate-binding module (CBM32) at the C-terminus and a lipoprotein signal peptide (Fig. 2A). The combination of a protease and glycosidases within the same PUL suggests that this system may target both the protein and glycan moieties of mucins.

### The mucin-active peptidase BT4244-M60L is surface-located and cleaves the *O*-glycosylated peptide linker of immunoglobulin A1 (IgA₁)

A classical functional PUL paradigm based on the mechanisms of several previously characterised Sus-like systems depicts a key endo-acting enzyme located on the cell surface which cleaves the target polymer into small oligomers (with degree of polymerisation of ~5-15 for the systems that have been studied) for import by the cognate SusC/SusD-like pair[8,14,25–27,45,46]. The presence of a predicted type II lipoprotein signal peptide and a CBM32 domain in BT4244-M60L are indicative of a possible cell surface association[32] (Fig. 2A). To investigate the expression and cellular location of native BT4244-M60L, polyclonal antibodies raised against the recombinant M60-like catalytic domain of BT4244-M60L were used to probe whole extracts of *B. theta* cells grown on glucose or PGMII following western blotting. The results showed detection of a protein band with a size of ~95 kDa (Fig. 2B, C), matching the expected size of BT4244-M60L. This band was absent in both glucose grown *B. theta* cells and deletion mutants lacking BT4244-M60L (ΔBT4244) (Fig. 2B, C). In another western blotting experiment with recombinant forms of the M60-like domains of BT4244-M60L homologues in *B. theta*, the polyclonal antibodies were shown to be highly specific for the M60-like domain of BT4244-M60L (Supplemental Fig. 1A). The cell surface location of the BT4244-M60L was further confirmed through immunofluorescence microscopy on fixed Wt and ΔBT4244 cells grown on PGMII (Fig. 2D).

We and others previously showed that BT4244-M60L is active in vitro on animal mucins as well as a range of *O*-glycosylated mucin domain proteins and model glycopeptides with a preference for Tn or Core 1 structures as the main cut sites[32–36]. However secretory IgA₁, like intestinal mucins, is also prominent in the gut and to our knowledge the activity of BT4244-M60L against this *O*-glycosylated protein has not been tested. IgA is the most abundant immunoglobulin in mucosal secretions and shares several important features with mucins (Fig. 1A, Fig. 2E)[47,48]. Human secretory IgA consists of two types including IgA₁ and IgA₂ (Fig. 2E)[49]. Notably, the IgA₁ heavy chain (HC) contains a hinge region between its Fab and Fc regions which is absent in IgA₂ (Fig. 2E). The hinge region contains proline, threonine and serine repeat sequences (PTS linker) which are also *O*-glycosylated with Tn and T antigen structures (often sialylated) as in intestinal mucins (Fig. 2E)[49,50], thus making them potential targets for PUL BT4240-50 in vivo. Incubation of BT4244-M60L with the two secretory IgA isoforms and other glycosylated/non glycosylated general protease substrates (casein, gelatin, bovine serum albumin and CD44[37,51,52]), followed by SDS-PAGE/western blotting analyses with anti-IgA HC antibodies and a GalNAc-binding lectin from *Helix aspersa* showed prominent degradation of IgA₁, but not IgA₂ (which lacks the *O*-glycosylated PTS linker of IgA₁) or any of the other proteins tested (with the exception of possible slight degradation of *O*-glycosylated CD44; (Fig. 2F and Supplemental Fig. 1B). Overall, these data reveal that IgA₁ is a substrate for the

 

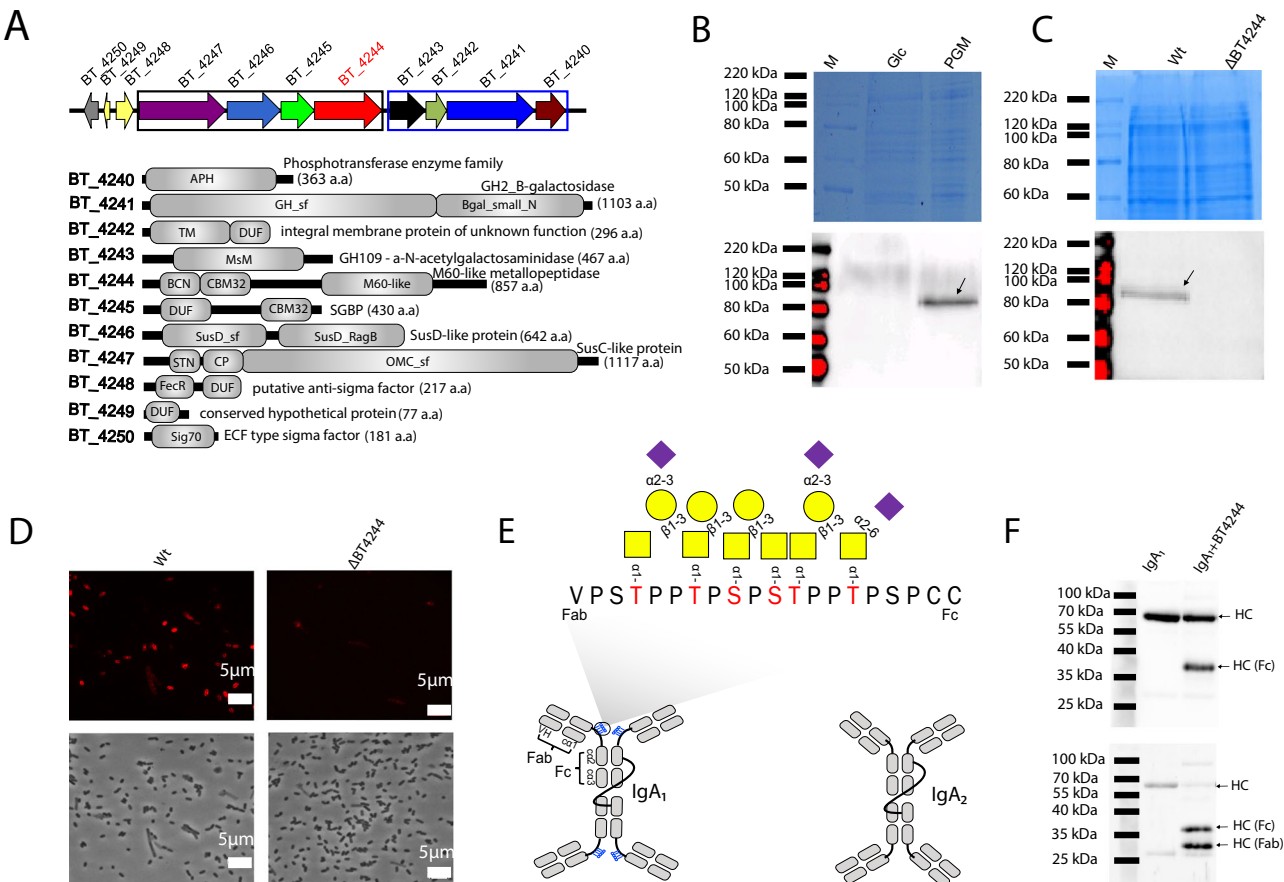

**Fig. 2 | The BT4240-50 locus and cellular location and activity of BT4244-M60L glycopeptidase. A** Top part of panel shows BT4240-50 locus with genetic context of BT4244-M60L and other PUL components. Predicted functions (Cazy, Pfam and KEGG annotations)[41,42] and modular architectures are shown below (drawn to scale; size of each ORF indicated brackets). The PUL is organised into two discrete operons (boxed in black and blue). The first operon (blue box) consisting BT4243-40 is expressed at a high basal level compared to the second operon (black box), containing genes encoding the hallmark SusC/D surface proteins (BT4246-7), BT4245 SGBP and BT4244-M60L, which are significantly upregulated during growth on mucins as the sole carbon source[17,32]. BT4244-M60L, is a mucin degrading metallopeptidase characterised previously[32]. **B** Induction and detection of BT4244-M60L following growth of *B. theta* in minimal medium containing either glucose (Glc) or PGMIII as sole carbon sources. Upper panel shows Coomassie blue staining of total cell lysate while bottom panel shows western blot detection of BT4244-M60L using polyclonal antibodies raised against the purified recombinant version of the M60-like catalytic domain of BT4244-M60L. **C** Detection of BT4244-M60L in wild type (Wt) or BT4244-M60L deletion strains (ΔBT4244). Upper panel shows total cell lysate stained with Coomassie; lower panel shows western blot probed with anti-BT4244-M60L antibodies. In (**B**, **C**) the arrows indicate a band of the predicted molecular weight of BT4244-M60L (~95 kDa). **D** Cellular localisation of BT4244-M60L by antibody immunofluorescence. Wt and ΔBT4244 cells were cultured in minimal medium containing mucin (PGMII) and probed with same antibodies in B/D followed by fluorescently labelled secondary antibodies (upper panels - see methods). Lower panels show phase contrast image of the same cells. **E** Model structure of IgA₁ and IgA₂ showing a *O*-glycosylated hinge peptide region in IgA₁. The IgA₁ linker contains 9 Ser and Thr residues that can be glycosylated but is most commonly glycosylated with 3-6 glycans at up to 6 main sites (shown in red) with a range of different sugar structures. Sugars are shown as per key in Fig. 1 **F** Proteolytic cleavage of IgA₁ by BT4244-M60L at the *O*-glycosylated hinge region. IgA₁ was incubated with 1 μM (final) BT4244-M60L in 20 mM Tris-HCl, pH 8.0, 100 mM NaCl for 16 h at 37 °C. Top and bottom panels show detection of intact and cleaved IgA₁ using anti-IgA HC antibodies and GalNAc-binding lectins, respectively. All experiments were performed at least three times with similar results.

BT4244-M60L glycopeptidase and that cleavage occurs at the *O*-glycosylated hinge region of the immunoglobulin. Additionally, we showed that growth of *B. theta* in minimal media containing PGMIII and IgA₁ leads to partial degradation of the immunoglobulin, suggesting that IgA₁ could be a target for BT4244-M60L in vivo (Supplemental Fig. 1C).

## CBM32 domains of BT4244-M60L and BT4245-SGBP recognise terminal galactose (Gal)-containing sugars

BT4244-M60L and BT4245-SGBP are lipoproteins each containing a CBM32 domain and hence are likely involved in carbohydrate binding interactions at the cell surface (Fig. 2A). BT4244-M60L also contains an N-terminal BACON domain with a yet to be defined function. A carbohydrate binding function was predicted for BACON domains based on bioinformatic analyses but to date this has not been experimentally demonstrated[53]. More recent structural and bioinformatic data

suggests BACON domains may act as linkers involved in positioning their attached C-terminal catalytic domains on the cell surface as part of their cognate SusC/D outer membrane utilisomes[25,45,53]. To gain more insights into the mucin structures targeted by BT4244-M60L and BT4245-SGBP, the ligand specificities of various modules were analysed by isothermal titration calorimetry (ITC) (Table S1; Supplemental Fig. 2A). When the BACON and BT4244-CBM32 were expressed individually and tested against various mucin derived mono- and disaccharides, it was observed that the CBM32, but not the BACON domain, was capable of binding sugars (Table S1), consistent with a linker/spacer role earlier proposed for the BACON domain in the context of the utilisome[25,45]. BT4244-CBM32 bound to GalNAc and Gal, as well as lactose and core 1 disaccharides, but not other sugars tested. Attempts to recombinantly express the CBM32 alone of BT4245-SGBP failed, however the full-length BT4245-SGBP, which was successfully expressed, displayed similar ligand specificity to the BT4244-CBM32,

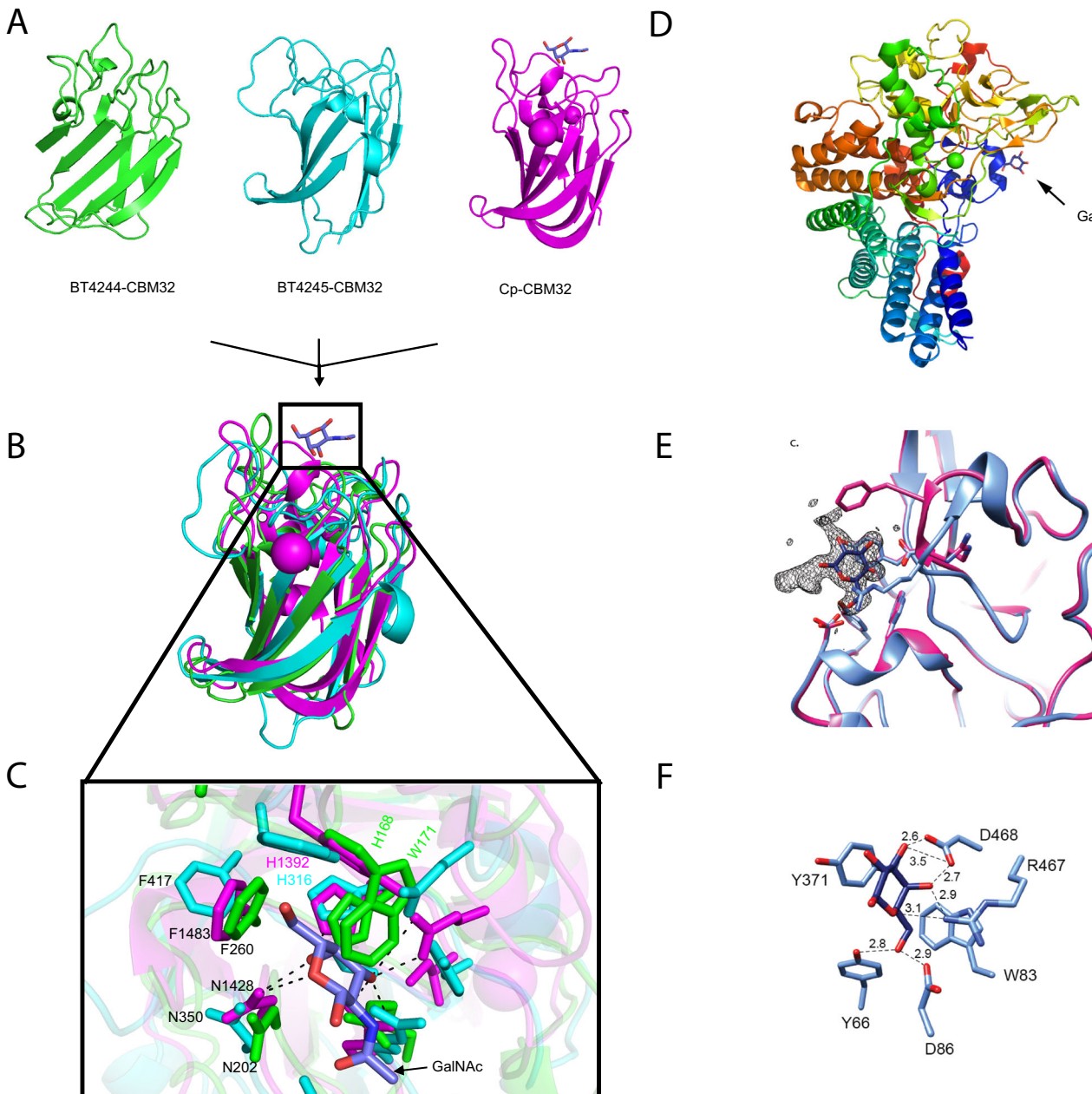

**Fig. 3 | Binding site comparison of CBM32 domains from various PUL BT4240-50 encoded proteins and the crystal structure of BT4246 SusD-like protein.**
**A** The BT4240-50 PUL encodes two proteins possessing CBM32 domains (BT4244-M60L and BT4245 SGBP). AlphaFold structures of both domains are compared with the crystal structure of the GalNAc binding CBM32 from a *C. perfringens* GH89 bound to GalNAc (CpCBM32-5, PDB: 4AAX, GalNAc shown in blue sticks).
**B** Alignment of all three CBM32 structures showing location of GalNAc binding site in CpCBM32 (GalNAc shown in blue sticks). **C** Binding site comparison from structural alignment in B showing residues involved in recognition of GalNAc in CpCBM32 and their equivalents in BT4244-CBM32 and BT4245-CBM32. Residues that were mutated in BT4244 include H168 and N202. Hydrogen bonds between the sugar and CpCBM32 are shown as dotted lines. Although very similar, the shape of the binding pocket maybe slightly different in the BT4244-CBM32 protein

considering the very close proximity of W171 to the CpCBM32 ligand, although this could also be an incorrectly modelled rotamer of this residue in the AlphaFold structure of BT4244-CBM32. **D** Crystal structure of BT4246-SusD in complex with a terminal Gal from a mucin *O*-glycan oligosaccharide. The protein backbone is colour-ramped blue to red from the N- to C-terminus, and a single Ca²⁺ is shown as a green sphere. Gal is displayed as dark blue sticks. **E** Omit map displaying the fo-fc density (δ = 3.0) for Gal in black mesh. Additional density can be seen extending off the O1 and O2 of Gal, but this could not be definitively assigned and did not interact with protein. The coordinates of the ligand-bound (blue) and SelenoMet apo (pink) structure are overlaid to demonstrate the plasticity of the loop defined by residues 462–467. **F** Zoomed-in view of Gal binding site pocket of BT4246 with sugar coordinating residues displayed as sticks. Dashed lines indicate potential hydrogen-bonding interactions within 3.5 Å.

but with ~3-fold higher affinity for GalNAc (Table S1; Supplemental Fig. 2A). Structural alignment of the CBM32 domains from BT4244-M60L, BT4245-SGBP (both AlphaFold models) and the well-characterised GalNAc-binding CBM32 domain of CpGH89 *C. perfringens* GH89 enzyme (CpGH89CBM32-5, PDB: 4AAX) showed spatial conservation of key residues implicated in ligand (GalNAc) recognition

in CpGH89CBM32-5 including H1392, N1428 and F1483, corresponding to residues N352, H316 and F417 in BT4245-SGBP and H168, N202 and F260 in BT4244-CBM32 (Fig. 3A–C). H1392 and N1428 make polar hydrogen bond interactions with the GalNAc ligand[54] in agreement with data from site-directed mutagenesis and subsequent ITC analyses which showed that the equivalent residues in BT4244-CBM32 (H168,

N202) are essential for binding to GalNAc (Fig. 3C, Supplemental Fig. 2B). The shared specificity of BT4244-M60L and BT4245-SGBP for a terminal Gal-configured sugar is consistent with the two proteins binding O-linked glycans on the surface of B. theta via their CBM32 domains. Notably, an AlphaFold model of the full length BT4244-M60L protein shows that the binding site of the CBM32 domain is facing the active site of the M60-like domain, supporting a role for the CBM in either substrate targeting or positioning, or both (Supplemental Fig. 3A).

### The PUL-encoded SusD-like BT4246 binds mucin-derived β-linked Gal disaccharides

SusD-like proteins are typically surface located lipoproteins that bind specific glycan fragments generated on the surface of the cell by their PUL-encoded endo-acting enzyme and deliver these to their partner SusC-like proteins for active import into the periplasm[14,17,22,25–27,45]. Understanding the glycan specificity of the PUL encoded BT4246-SusD-like could therefore provide further insight into the mucin components targeted by PUL BT4240-50. The ITC data showed that BT4246-SusD-like displayed quantifiable binding to the disaccharide sugars galacto-N-biose (core 1; Galβ1,3 GalNAc) and lacto-N-biose (LNB; Galβ1,3GlcNAc) (with a >3-fold preference for the core 1 structure) but not the monosaccharides components of the same sugars, β1,4-linked disaccharides (lactose and LacNAc) or α-linked disaccharides (Galα1,3Gal or GalNAcα1,3GalNAc) (Table S1; Supplemental Fig. 2A). These data suggest BT4246-SusD-like preferentially targets β1,3-linked Gal sugars.

### Structure of BT4246-SusD-like bound to a mucin-derived oligosaccharide reveals recognition of a terminal Gal

The structure of BT4246-SusD-like (PDB: 5CJZ), solved to 1.8 Å, revealed the protein adopts a typical SusD-like helical fold (Fig. 3D)[46,55]. To investigate the ligand binding site of BT4246-SusD-like, the native protein crystals were soaked in a 4% solution of O-glycan oligosaccharides purified from PGMIII[17]. In the resulting structure, unambiguous electron density was observed for a β-linked Gal, suggesting that this terminal moiety was preferentially 'selected' from the heterogeneous mixture (Fig. 3E). Gal could only be modelled with the non-reducing end of the sugar wedged into the shallow surface pocket that is within the known ligand-binding region of other SusD-like proteins (Fig. 3D, E, Supplemental Fig. 3B and Table S2)[8]. Additional density was observed extending off the O1 and O2 of the Gal in BT4246 but could not be definitively assigned and did not directly interact with the protein (Fig. 3E). This mode of recognition may be a particularly effective way to import a variety of peptide-linked glycans such as core 1 structures that feature a terminal Gal (for example see Supplemental Fig. 4). The Gal is recognised via hydrophobic stacking with Y371 and the side chains of residues Y66, D86, R467, and D468 are proximal for hydrogen-bonding (Fig. 3F). Glucose cannot be accommodated in this site as the O4 would sterically clash with W83, and the hydrogen bond between the O4 and the carboxylate side chain of D468 would be lost (Fig. 3F). However, the selectivity of the protein for β-Gal is unclear, as there is no obvious steric hindrance for binding of α-linked Gal.

While there is no significant conformational change that occurred upon ligand-binding, we observed differences in orientation of the loop defined by residues 462–467 between the native protein crystals (space group P622) and that of the Selenomethionine (SelenoMet)-substituted protein crystals (space group C2221) (Fig. 3E). In the SelenoMet-BT4246-SusD-like structure, F465 partially occludes the binding site, while R467, which is proximal to the Gal O1 and O5 atoms in the native crystals, is oriented out of the binding site. These changes suggest that there is some inherent plasticity of the protein surrounding the binding pocket that may facilitate ligand recognition. Of resolved SusD-like structures, BT4246 is most similar to BT3013 (PDB 7BLL, Z-score = 39.0, 2.4 Å rmsd Cα, 34% identity), which is within an

uncharacterised PUL that encodes a predicted M60-like peptidase (BT3015), but no CAZymes (Supplemental Fig. 5). While the encoded proteins have not been characterised, the PUL is upregulated during B. theta growth in vitro on a preparation of porcine mucosal glycans[17]. Despite overall structural conservation with BT4246, BT3013 SusD-like does not share conservation of any of the residues within the ligand-binding pocket, including those that capture Gal in BT4246, suggesting a different ligand specificity (Supplemental Fig. 5).

### PUL BT4240-4250 encodes a β1,3-galactosidase and α-N-acetyl-galactosaminidase that target mucins and mucin-derived structures

To explore the activities of PUL-encoded glycoside hydrolases BT4241-GH2 and BT4243-GH109, recombinant forms of the proteins were assayed against a range of PNP-monosaccharides, mucins and mucin-derived disaccharides. BT4241-GH2 was shown to be β-galactosidase cleaving several β-linked Gal-containing substrates with a ~100-fold preference for 1,3 over 1,4-linked Gal, an unusual specificity for GH2 β-galactosidases which commonly prefer the latter linkage (Fig. 4A and Table S3)[23]. The highest activity observed was against galacto-N-biose (core 1; Galβ1,3GalNAc), a structure that is known to be resistant to hydrolysis by many intestinal bacteria[11] followed by lacto-N-biose (Galβ1,3GlcNAc) and the non-mucin derived β1,3 galactobiose, revealing ~3-6-fold preference for GalNAc over either GlcNAc or Gal at the +1 sub-site (Table S3). BT4241-GH2 was also able to release galactosyl residues from mucins such as porcine gastric mucin II (PGMII), but only after pre-treatment with an α1,2-fucosidase to remove the fucosyl residues that typically cap the Gal sugars in these glycoproteins (Fig. 4B)[56]. The enzyme displayed no activity however against the β1,3GlcNAc linkage in core 3 disaccharide (GlcNAcβ1,3GalNAc), a prominent core structure found in colonic mucin in particular MUC2[57] (Fig. 4C).

Analysis of the specificity of the BT4243-GH109 enzyme revealed it to be an exo-acting α-N-acetylgalactosaminidase (α-GalNAc'ase) capable of hydrolysing the α1-O-linked glycosidic bond between GalNAc and serine in Tn antigen as well as PNP-α-GalNAc (Fig. 4D and Table S3), the former an activity not previously described for this family. Notably, BT4243-GH109 was inactive on T-antigen (Galβ1,3GalNAcα-1-O-Ser) unless pre-treated with the PUL-encoded BT4241-GH2 β-galactosidase (Fig. 4D). The BT4243-GH109 enzyme was also active on bovine submaxillary mucin (BSM) with the amount of GalNAc released increasing significantly following pre-treatment with a sialidase enzyme (Fig. 4E). This is in agreement with a previous report showing that much of the α-GalNAc in BSM is capped with sialic acids[58]. Additionally, BT4243-GH109 was active on the Forsmann disaccharide sugar (GalNacα1,3GalNAc) and the peripheral BgA, both of which contain a terminal α-1,3GalNAc linkage (Fig. 4F, Fig. 5)[57,59,60].

Other GH families with α-GalNAc'ase activity include GH27, GH31, GH36, GH101, and GH129 (CAZy.org). Several genes encoding these GH families except GH101 and GH129 are annotated in the genome of B. theta (CAZy.org). Recently, a GH31 α-GalNAc'ase from B. caccae (BACACC_01242) that targets the core GalNAc in O-glycans from fetuin and a glycopeptide, was reported[61] and a homologue of the B. caccae enzyme was identified in B. theta (BT3169) sharing significant sequence and domain features (75% ID) (Supplemental Fig. 6A–C). BT3169-GH31 has also previously been implicated in O-glycan processing[55]. Analysis of BT3169-GH31 activity against Tn-antigen, BgA and desialyated BSM showed that the enzyme also exhibits α-GalNAc'ase activity but is only active against the mucin and not Tn or BgA (Fig. 5). The lack of activity of BT3169-GH31 vs BgA is similar to the B. caccae enzyme and a homologue from Enterococcus faecalis[62], but different to BT4243-GH109, which was active against all three substrates (Fig. 5). Notably however, BT3169-GH31 appeared to display higher activity than BT4243-GH109 against BSM, suggesting that the GH31 may play a more significant role in deglycosylating intact mucins than BT4243, which seems to prefer smaller mucin breakdown products, such as Tn

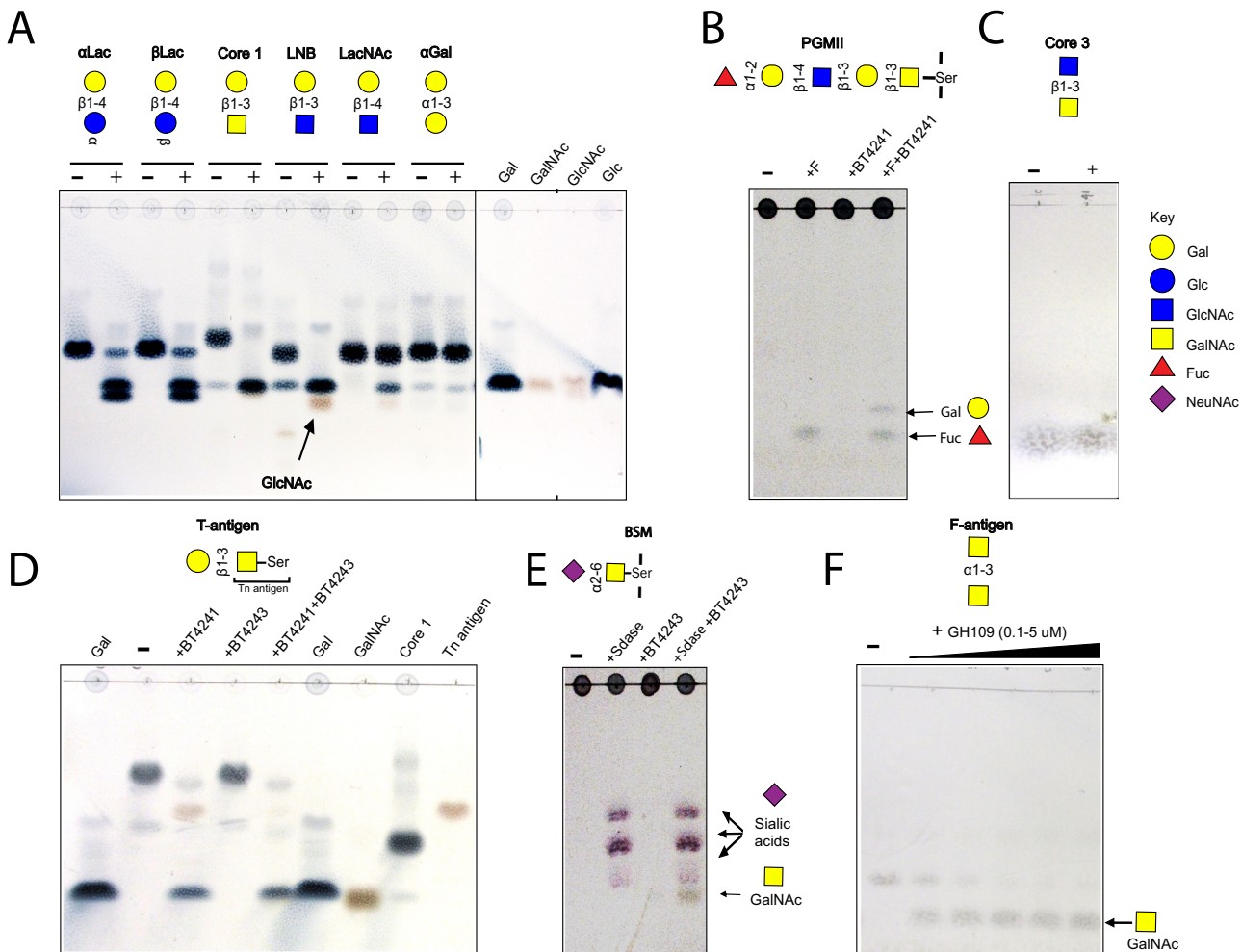

**Fig. 4 | Activity of PUL BT4240-4250 encoded glycoside hydrolases against mucins and mucin-derived disaccharides. A** BT4241-GH2 activity against various mucin-derived disaccharides. Lane (-) represents substrate alone without enzyme and (+) with added enzyme. Note that when the core 1 sugar is cleaved, the released Gal co-migrates with GalNAc and the Gal masks the GalNAc. In addition, GlcNAc co-migrates with GalNAc when run as a standard, but runs slightly below Gal in the LNB and LacNAc assay lanes. **B** Activity of BT4241-GH2 against PGMIII mucin with and without pretreatment with an α1,2-fucosidase (F). The majority of PGMIII glycans contain terminal Fucα1-2-Galβ1-4 GlcNAc structures as shown above (example PGM sugar chain shown). **C** Treatment of core 3 with BT4241-GH2 (+ indicates with enzyme). No activity was detected against this substrate indicating the enzyme is does not cleave β1,3-linked GlcNAc. **D** Activity of BT4241-GH2 and BT4243-GH109 against T-antigen (Galβ1,3GalNAcα1-Ser). BT4241-GH2 releases Gal and Tn antigen

(GalNAcα1-Ser) from T-antigen. BT4243-GH109 is then able to release GalNAc from Tn antigen (GalNAcα1-Ser). As in A above, the GalNAc spot gets masked on the TLC by Gal, but the activity can be seen by the disappearance of the Tn antigen spot. Core 1 disaccharide (Galβ1,3GalNAc) is also shown as a standard. **E** Activity of BT4243-GH109 against BSM. The majority of BSM glycans contain sialylated Tn antigen. After desialylation with a sialidase (Sdase), BT4243-GH109 cleaves the uncapped core GalNAc from the mucin. **F** BT4243-GH109 cleaves F-antigen (core 5) disaccharide showing concentration dependent cleavage of the substrate. Reactions were generally set-up to contain 5-10 mM of disaccharides or 4 mg/ml mucins and 1 μM of each enzyme except where indicated in 20 mM Tris-HCl, pH 7.5 and incubated at 37 °C for 1 h (except for PGM, BSM, core 3 and T-antigen incubated for 12 h) before spotting on TLC. All experiments were performed at least three times with similar results.

antigen, as well as the terminal GalNAc decorations found in BgA. Overall, these data show that *B. theta* encodes at least two different enzymes capable of removing core GalNAc from mucins as well as a β1,3-galactosidase capable of targeting mucin core 1 structures.

## BT4240 is a GalNAc specific kinase

BT4240 is annotated as a phosphotransferase containing an APH (aminoglycoside phosphotransferase family (PF01636)) motif; Fig. 2A) and displays 39% identity to an N-acetylhexosamine kinase from *Bifidobacterium longum*, BL1642[63,64]. To investigate the activity of BT4240, a recombinant form of the protein was tested for its ability to phosphorylate a range of monosaccharides and hydroxy-amino acids found in *O*-glycans. The data revealed that BT4240 was indeed an N-acetylhexosamine kinase with a strong preference (~275-fold) for GalNAc over GlcNAc, unlike the *B. longum* enzyme that displays similar activity against both sugars (Fig. 6A, Table S3). BT4240-kinase showed

no activity on any of the other monosaccharides or amino acids tested (Fig. 6A). Production of the phosphorylated sugars was also monitored by thin layer chromatography (TLC) (Fig. 6B). To determine if phosphorylation of GalNAc occurred before or after release of the sugar from the peptide, the activity of BT4240-kinase against the Tn antigen (GalNAcα1-Ser) was measured in the presence and absence of BT4243-GH109 α-GalNAc'ase (Fig. 6C, D). The data showed that the rate of phosphorylation of GalNAc increased dramatically (monitored by the decrease in $A_{340nm}$) following the addition of the BT4243-GH109 to the assay, indicating that significant phosphorylation of the sugar occurs only after its release from the Tn-antigen. This suggested that either BT4240-kinase preferentially phosphorylates the sugar at position O1 which is linked to Ser (or Thr) in the Tn antigen, or the presence of the amino acid impacts phosphorylation at other sites including at position O6, which is a common phosphorylation site of sugars. To investigate this further, we analysed the product of GlcNAc

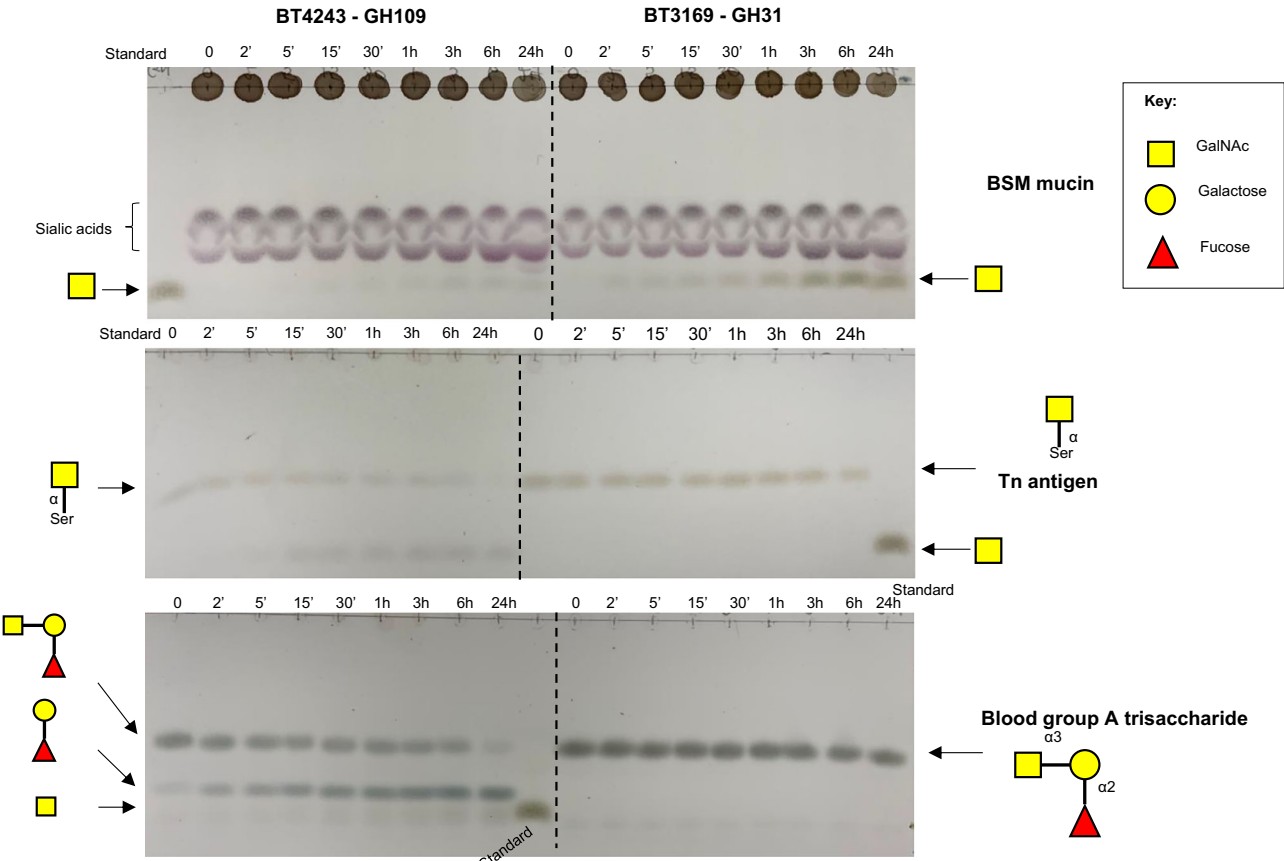

**Fig. 5 | Comparison of BT4243-GH109 and BT3169-GH31 activities against mucin derived α-GalNAc containing structures.** Potential substrates (structures shown on the right of each TLC) were incubated with the same concentration of the two α-GalNAc'ases (1 μM final) in 20 mM Tris-HCl, pH 7.5 at 37 °C and samples collected at various time points and analysed by TLC. BT4243-GH109 was active against all three substrates, but with a strong preference for BgA, while BT3169-GH31 was only active against BSM (mainly Tn antigen when de-sialylated), although appeared to show higher activity vs the mucin than BT4243-GH109. NB. BSM was pre-treated with *B. theta* sialidase (BT0455) to uncap core GalNAc residues. All experiments were performed at least three times with similar results.

phosphorylation by BT4240-kinase by high performance anion exchange chromatography with pulsed amperometric detection (HPAEC-PAD) and compared the product to commercially available GlcNAc-1-phosphate (G1P) and GlcNAc-6-phosphate (G16P) standards as phosphorylated GalNAc standards were not available. Both G1P and G6P had different retention times by HPAEC-PAD which confirmed production of G1P but not G6P following treatment of GlcNAc with BT4240-kinase (Fig. 6E). Further support for O1 phosphorylation is provided by a comparison of the AlphaFold model of the *B. theta* kinase with the *B. longum* HexNAc kinase, BL1642 which has been experimentally and structurally shown to phosphorylate at O1 of Gal-NAc and GlcNAc (Supplemental Fig. 7)[63,64]. The sequence alignment and structural overlay reveal that most of the residues comprising the sugar and ATP binding sites in BL1642 are conserved in BT4240-kinase including the key catalytic residues involved in the phosphorylation at O1 (Supplemental Fig. 7B–E). Furthermore, the position of the GalNAc from BL1642-kinase when overlaid onto the BT4240-kinase model is such that phosphorylation of O6 of the sugar is precluded, further supporting O1 as the site of phosphorylation by BT4240-kinase. Notably, there are no additional homologues of BT4240-kinase in the *B. theta* genome, suggesting that it may be the only GalNAc kinase encoded by the bacterium.

### Concerted action of PUL BT4240-50 encoded GHs provides insight into BT4244-M60L activity against IgA₁

The activities obtained so far for the various PUL BT4240-50 binding proteins and enzymes are highly complementary and highlight a sequential substrate degradation mode culminating in the release of

the core mucin *O*-glycan sugars GalNAc and Gal from glycoprotein substrates such as imported mucin-derived glycopeptides. As the targets for these enzymes - including the Galβ1,3GalNAc-peptide (i.e. core 1 or T-antigen) are present on a natural substrate such as IgA₁ (Fig. 7A), this offered a useful opportunity to not only assess the impact of *O*-glycosylation on the activity of the BT4244-M60L glycopeptidase, but also gain an insight into the hierarchical order in which the different PUL components work. To achieve this, the PUL encoded GHs and a broad-acting neuraminidase were used to sequentially de-glycosylate the antibody and investigate the effect of this on BT4144-M60L activity. The results, judging by the intensity of the released Fc fragment, showed an increase in the degradation of IgA₁ with increasing deglycosylation up to GalNAc (Fig. 7B, C). The staining with the anti-GalNAc lectin also increases in intensity with the removal of the sialic acid and Gal to leave only Tn antigen, but the lectin staining is lost on addition of BT4243-GH109, demonstrating removal of all remaining core GalNAc by the α-GalNAc'ase (Fig. 7B). Notably, the complete deglycosylation of IgA₁ by addition of BT4243-GH109, led to a significant reduction in the activity of BT4244-M60L against the immunoglobulin, as seen by the loss in intensity of the released Fc band (Fig. 7B, C). These data are consistent with previous studies of BT4244-M60L against synthetic *O*-glycopeptide substrates and recombinant mucin domain glycoproteins, where it was demonstrated that the enzyme preferentially targets unsialylated *O*-glycopeptide sequences (containing GalNAc or Galβ1,3GalNAc side chains, i.e. Tn or T-antigen, respectively) and has very limited activity against fully unglycosylated peptide[34,33]. Overall, these data support the hypothesis that BT4244-M60L is able to cleave

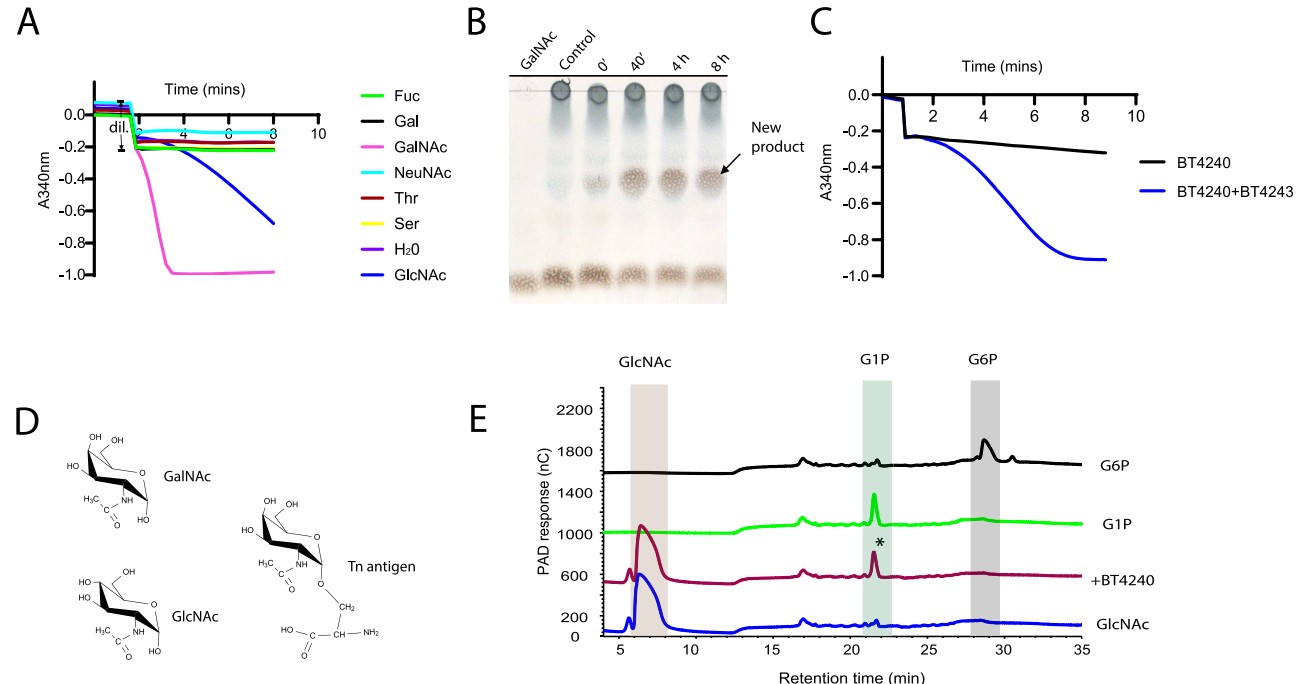

**Fig. 6 | Activity of BT4240-kinase against various substrates. A** Kinase activity was tested against various mucin derived monosaccharides and amino acids using a linked kinase assay that generates NADH following phosphorylation of the substrate leading to a decrease in absorbance at 340 nm ($A_{340}$)[98]. Region of curve annotated 'dil.' shows an initial decrease in $A_{340}$ due to dilution of the reaction mixture following addition of BT4240-kinase. A major decrease in $A_{340}$ was detected for GalNAc and GlcNAc substrates indicating kinase activity. **B** TLC data showing phosphorylation of GalNAc by BT4240-kinase and production of the new product (likely GalNAc-1-phosphate) and depletion of GalNAc substrate over time. Control lane is with no added enzyme **C** Activity of BT4240-kinase on Tn-antigen and BT4243-GalNAc'ase treated Tn-antigen. **D** Structures of some main substrates used to assess BT4240-kinase activity. GalNAc is strongly preferred over GlcNAc and the Tn antigen. **E** HPAEC-PAD analyses showing phosphorylation of GlcNAc by BT4240-kinase and comparison of the generated product with phosphorylated GlcNAc standards GlcNAc-1-phosphate (G1P) and GlcNAc-6-phosphate (G6P). The peak of the product from the activity of BT4240-kinase on GlcNAc (starred *) is seen to have a similar retention time as the G1P, confirming that the enzyme catalyses conversion of GlcNAc to G1P. All kinase experiments were performed at least three times with similar results (see additional replicate data and kinetic calculations or analyses in supplemental Table S3 and source data). Source data are provided as a Source Data file.

glycosylated PTS repeats of IgA₁ and preferentially targets T and Tn antigen structures.

### PUL BT4240-50 is important for competitive growth on mucins

To explore the importance of the PUL BT4240-50 in mucin metabolism, a deletion (or KO) mutant lacking the entire PUL BT4240-50 (ΔBT4240-50) was created and assessed for its ability to utilise PGMIII. ΔBT4240-50 and the Wt strains grew at similar rates on glucose and in the early stages of growth on PGMIII (Fig. 8A, B). The mutant however displayed a small but reproducible reduction in maximal density at the later stages of growth on PGMIII, suggesting a component of the PGMIII was not available to the mutant strain (Fig. 8B). The growth defect was significantly more dramatic in competition experiments in vitro between the Wt and KO strains, where the KO was rapidly outcompeted when the carbon source was switched from glucose to PGMIII (Fig. 8C). The data thus demonstrate that PUL BT4240-50 is important for competitive access to at least one component of mammalian mucins. To assess what this might be, we tested growth of the mutant on two of the main constituent monosaccharides of the mucin, Gal and GalNAc (Fig. 8D, E). Growth experiments with the monosaccharides produced an interesting result in that the PUL BT4240-50 deletion mutant completely lost its ability to grow on GalNAc but not Gal (Fig. 8D, E), indicating that one or more components of PUL BT4240-50 is important for growth on GalNAc. Interestingly, a BT4240-kinase deletion mutant ΔBT4240 completely failed to grow on GalNAc over a 20 h period compared to the Wt strain (Fig. 8F) and the ΔBT4240 mutant showed defective growth on PGMIII, similar to the levels observed for ΔBT4240-50 on the same substrate.

Complementation of the ΔBT4240-50 mutant with BT4240 restored its growth to near Wt levels (Fig. 8G, H). These data thus reveal that the major contribution of ΔBT4240-50 during growth on PGMIII is the metabolism of GalNAc by the kinase. This is in line with evidence showing redundancy in the activity of the other PUL BT4240-50 components such as the α-N-acetylgalactosaminidase activities observed for BT3169-GH31 and BT4243-GH109 which both release GalNAc from mucins (Fig. 5)[61] and the widespread redundancy of GH2 genes across mucin-inducible loci[19]. Additionally, a ΔBT4244 mutant did not show a significant growth defect on PGMIII and neither was it outcompeted in competition experiments (Supplemental Fig. 8) also consistent with the presence of several other predicted M60-like or other putative mucin-active proteases from different MEROPS families encoded by *B. theta* (Table S4). In the same light, deletion of the BT4242 gene encoding the putative cytoplasmic membrane sugar transporter did not affect growth on various substrates including Gal, GalNAc and PGMIII compared to the Wt strain (Supplemental Fig. 8) further indicating the likelihood of redundancy in PUL BT4240-50 functions, except for the BT4240-kinase.

### BT4240-kinase plays a central role in global GalNAc metabolism

Out of the myriad of genes (at least 100) and PULs induced during mucin metabolism and the high rate of redundancy in mucin-degrading gene families in *B. theta*[16,27,28], it was intriguing to observed that deletion of the BT4240-kinase alone was able to prevent growth of the organism on GalNAc. This led us to hypothesize that BT4240-kinase's role likely extends beyond mucin metabolism to other GalNAc-containing glycans. To test this, we cultured the

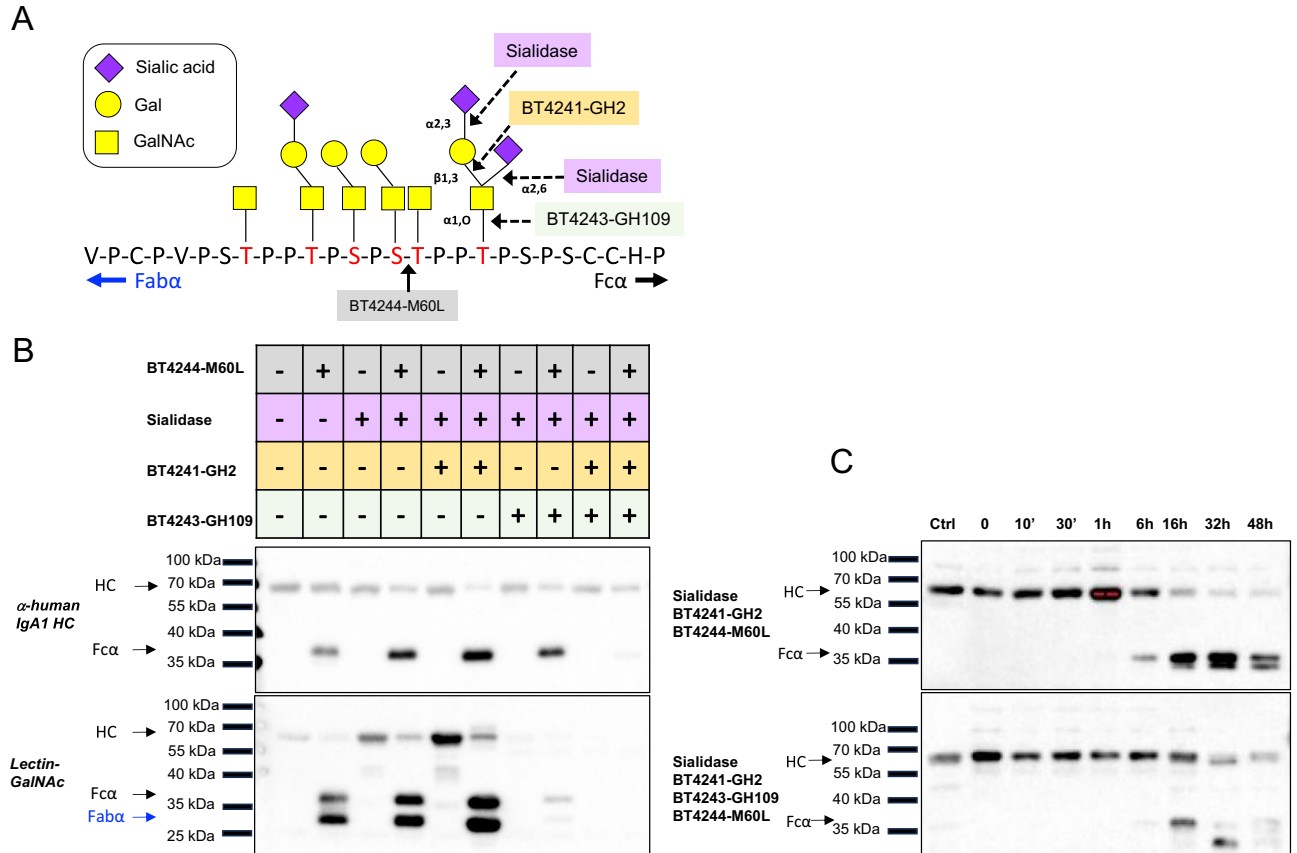

**Fig. 7 | Impact of glycosylation of IgA$_1$ on BT4244-M60L peptidase activity.**
**A** IgA$_1$ peptide hinge region showing *O*-glycans and target sites for PUL encoded glycoside hydrolases (BT4241-GH2 β1,3-galactosidase and BT4243-GH109 α-Gal-NAc'ase), sialidase and BT4244-M60L peptidase. The linker is shown partially desialylated to show the preferred substrate of BT4244-M60L. The IgA$_1$ linker can be glycosylated at up to 6 different sites (shown in red) with a range of different sugar structures[105,106] and an example glycoform is shown here. **B** Detection of IgA$_1$ breakdown products after digestion with various enzyme combinations. The upper panel shows the presence (+) or absence (-) of different enzymes in the reaction mixture with IgA$_1$. IgA was pretreated with sialidase (0.01 units), BT4242

galactosidase (0.5 μM final) or BT4243 α-GalNAc'ase (1 μM final) in 20 mM Tris-HCl, pH 8.0, 100 mM NaCl 16 h, before incubation with BT4244-M60L glycopeptidase (0.7 μM final) for a further 16 h at 37 °C. Reactions were run on an SDS-PAGE gel, blotted and proteins detected with either anti-IgA$_1$ heavy chain (HC) antibodies (middle panel) or a GalNAc specific lectin (lower panel). **C** Time course of BT4244-M60L vs partially (top panel) or fully (bottom panel) deglycosylated IgA1. Blots were stained with anti-HC antibodies. Reactions were carried out in the same conditions and the same concentrations of GHs and glycopeptidase as in panel B. IgA degradation experiments were performed at least three times with similar results.

ΔBT4240 strain in the presence of a non-mucin, but GalNAc-containing glycan substrate, chondroitin sulphate (CS) (Fig. 8I), which also happened to be a prominent nutrient source for the human gut microbiota targeted by *B. theta*[17,65,66]. The mechanism of the CS PUL has been described previously, revealing that GalNAc and 5-keto, 4-deoxyuronate are major end products of the degradative process[65]. We thus anticipated that if BT4240 metabolises GalNAc from CS, the BT4240 mutant would show a significant growth defect commensurate with the proportion of GalNAc in the CS, which is about 50%. This was confirmed whereby the ΔBT4240 strain was only able to grow to about 50% of the level of the Wt on CS at various stages of growth (Fig. 8I). The results confirm that BT4240-kinase processes GalNAc from other glycan sources like CS and likely represents a central metabolic hub for GalNAc processing in *B. theta*.

### Cellular localisation of PUL BT4240-50 proteins provides insights into the pathway for mucin breakdown in *B. theta*

To provide a better understanding of the mechanism of action of PUL BT4240-50, the cellular locations of five PUL components including BT4240-kinase, BT4241-GH2, BT4243-GH109, BT4244-M60L and BT4245-SGBP were investigated. To do this, the native proteins in *B. theta* were FLAG®-tagged at their C-termini and their locations tracked in various cellular extracts (prepared by centrifugal fractionation of

whole cell lysates/sonicate of *B. theta* cells) by western blotting with antibodies against the FLAG® peptide. The BT4244-M60L protein which was earlier confirmed to be surface localised (Fig. 2B–D) was also FLAG®-tagged and used as control. Analyses of the lysis product of all five engineered strains showed detection of all the FLAG®-tagged proteins (Supplemental Fig. 9). The BT4240-kinase was solely detected in the soluble fraction but not the cell membrane fractions (CMF). BT4241-GH2 was detected in both cell membrane and soluble fractions but were more enriched in the soluble fraction. BT4243-GH109 had a very similar profile to BT4241-GH2. As expected, and in contrast to BT4240-kinase, BT4244-M60L was mainly enriched in the cell membrane fraction with a reduced amount in the soluble fraction. The same profile was also observed for BT4245-SGBP suggesting that the protein is also cell surface-localised as predicted for a SBGP. BT4240-kinases's profile is in line with its in silico cytoplasmic prediction. It is also interesting that BT4240-kinase was almost completely absent from the insoluble CMF fraction as opposed to the proteins predicted to be periplasmic, which could still be detected in small amounts in the CMF fractions. It is possible that the small amounts of periplasmic proteins detected in the CMF are proteins caught in transit through the inner membrane to the periplasmic space and could explain the complete absence of BT4240-kinase observed in the CMF since it doesn't have to go through the inner membrane. Combining the cellular location and

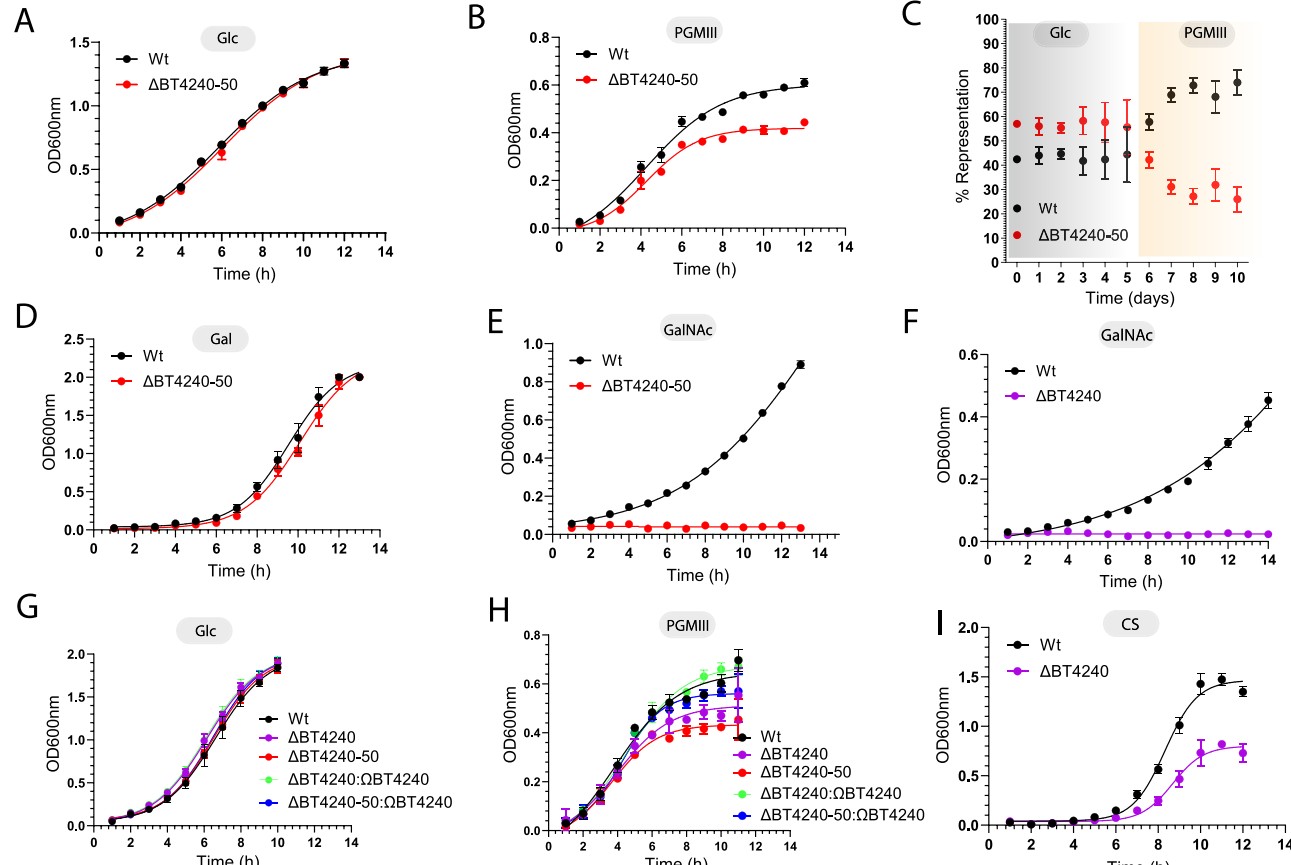

**Fig. 8 | Growth and in vitro competition experiments with *B. theta* Wt and mutants on mucins, GalNAc and chondroitin sulphate. A–I** show growth profiles produced by monitoring $OD_{600}$ of each culture of deletion and complementation mutants in minimal media (MM) with different monosaccharides, PGMII or chondroitin sulphate (CS) as the sole carbon source. **C** Shows competition experiments whereby tagged *B. theta* Wt and ΔBT4240-50 mutant cells were co-cultured in MM containing Glc and then switched to MM containing PGMIII as the sole carbon source. The percentage representation of each strain was quantified by qPCR analysis. Error bars indicate standard deviations of three independent cultures. Deletion and complementation mutants are represented by Δ and Ω, respectively. Data points of various measurements are presented as mean values +/- standard deviation and were obtained from triplicate experiments. Source data are provided as a Source Data file.

activity data for each protein, a model of the function of the PUL was proposed (Fig. 9). In this model, naturally occurring short mucin chains (e.g., Tn and core-1-containing chains), or those generated by the activity of surface glycosidases such as GH16 endo-*O*-glycanases (from *B. theta* and other members of the mucolytic microbiota[14]) are bound on the surface of the cell by the surface proteins BT4245-SGBP and BT4244-M60L through their respective CBM32 domains. The mucin glycoprotein backbone is cleaved endolytically by BT4244-M60L, and the resulting short glycopeptides imported into the cell through the PUL encoded SusC/D transporter. While recognition of terminal Gal-containing glycopeptides by BT4246-SusD like could provide a mechanism for glycopeptide uptake that does not require binding to the variable peptide sequence, additional interactions between the peptide and the SusC or SusD may also facilitate the selection of pre-ferred glycopeptide substrates (Supplemental Fig. 4). Periplasmic enzymes BT4241-GH2 and BT4243-GH109 sequentially cleave Galβ1,3 and GalNAc1-*O* linkages, respectively, from the imported glycopeptides, releasing Gal and GalNAc. The peptide-attached GalNAc can also be released by other periplasmic GalNAc'ases including the BT3169-GH31. Released GalNAc residues from various sources (core and terminal decorations and chondroitin sulphates[65]) are imported into the cytoplasm for phosphorylation by BT4240-kinase and used for the generation of energy and carbon fixation, promoting growth (Fig. 9). While the data presented here supports this model of mucin process-ing by PUL 4240-50 proteins, we do not show direct evidence of

glycopeptide binding by the SusD and so it is possible that alternate pathways for mucin breakdown by the PUL components exist. Fur-thermore, it is not clear what happens to the internalised peptide once it has been deglycosylated. It may be there are exo-acting periplasmic peptidases that release amino acids for uptake and metabolism, or alternately *B. theta* may secrete the deglycosylated peptide back into the intestinal milieu for other members of the microbiota to cross feed on.

## Origin and distribution of PUL BT4240-50 across *Bacteroides spp.* and metagenomes

To investigate the distribution of the BT4240-50 PUL across *Bacter-oides* spp., a combination of BlastP searches with all proteins encoded by the PUL was used as query and gene neighbourhood analyses identified identical PULs to BT4240-50 in 13 *B. theta* annotated gen-omes at the IMG/M database[67], which include isolates from domes-ticated animals[68] in addition to humans (Supplemental Fig. 10). Consistent with the conservation of the PUL BT4240-50 between human and animals *B. theta* strains, all 41 *B. theta* reconstructed gen-omes from human faecal metagenomes[69] also contained the PUL. These analyses also identified BT4240-50-like PULs in the annotated genomes from *Bacteroides faecis* and *B. caccae* (Supplemental Fig. 11). Scanning 287 human faecal metagenome sequence data identified the PUL BT4240-50 in 54% of the samples, a similar level to the *B. theta* fungal α-mannan PUL2 BT3773-92 (56%) (Supplemental Fig. 12)[24]. The

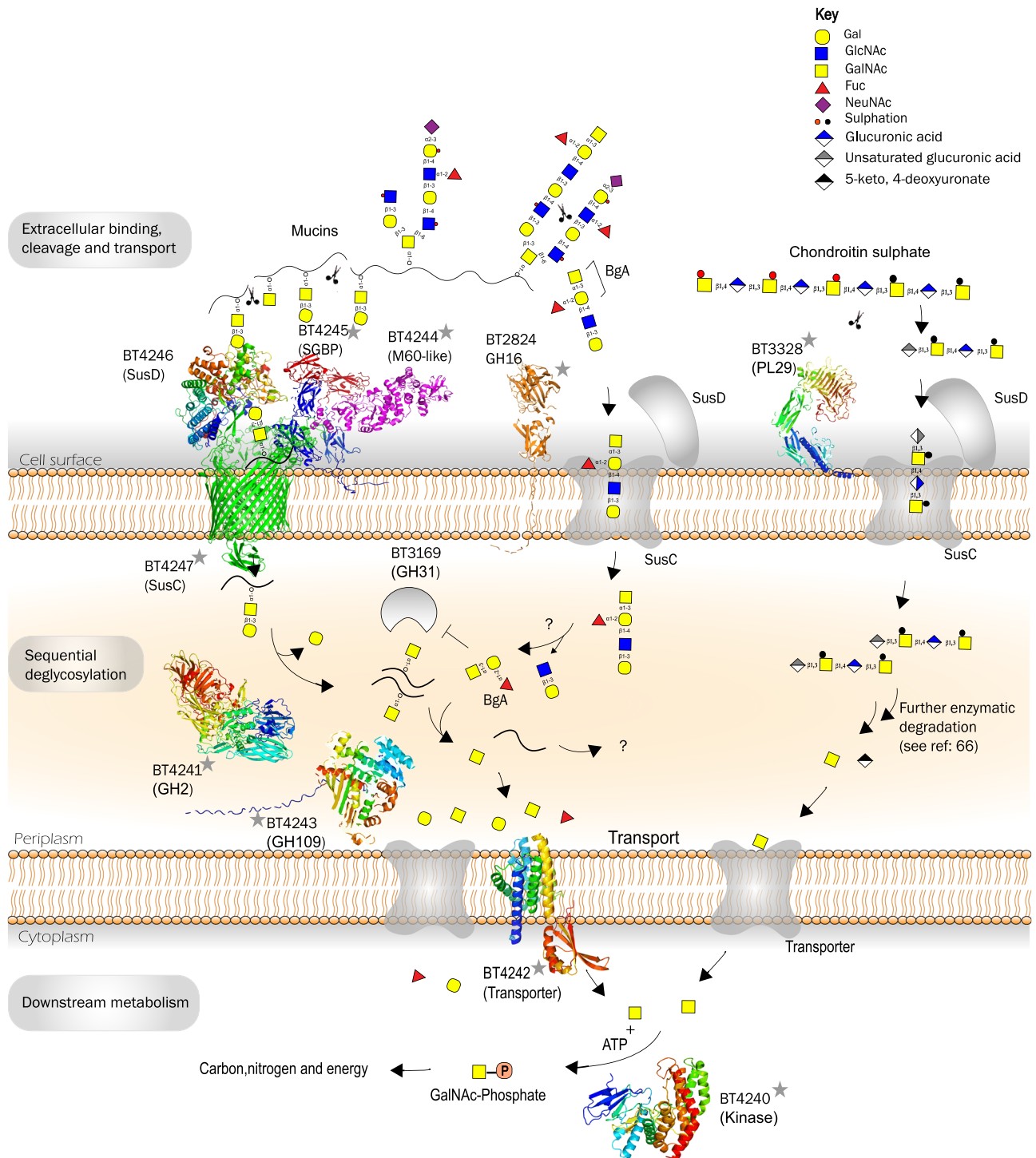

**Fig. 9 | A model pathway for the breakdown of mucin *O*-glycoproteins by PUL BT4240-50.** Terminal Gal- and GalNAc-containing mucin *O*-glycans are bound at the cell surface by CBM32s from BT4245-SGBP and BT4244-M60L peptidase. The BT4244-M60L peptidase endolytically cleaves the glycosylated mucin peptide backbone at Tn and core 1 structures, releasing glycopeptides that are bound by the SusC/D transporter proteins BT4246/BT4247 and imported across the outer membrane. GH16 endo-*O*-glycanases (e.g. BT2824) also cleave the mucins endolytically at the cell surface, but act on the oligosaccharide side chains, thus likely cleaving prior to the BT4244 endopeptidase[14]. Once in the periplasm the glycopeptides (core 1 and Tn) are deglycosylated by the sequential action of BT4241-GH2 β1,3-galactosidase and BT4243-GH109 and BT3169-GH31 α-N-acetyl-galactosaminidases. The fate of the deglycosylated peptide is not known, while the Gal and GalNAc released are then transported into the cytoplasm by yet to be defined or redundant transporters including BT4242-MFS, where the GalNAc is phosphorylated by the BT4240-kinase for downstream metabolism. GalNAc released from other structures e.g. mucin BgA decorations by BT4243-GH109, as well as from breakdown of other glycans such as chondroitin sulphate[66,67] is also processed by BT4240-kinase. Note that, whilst this model is largely supported by experimental data from our work, it is possible that these systems employ alternative mechanisms and can process other glycan/glycoprotein targets yet to be identified. Starred protein structures were generated using AlphaFold while the BT4246-SusD-like is the crystal structure bound to galactose from this study (PDB 5CJZ).

combined distribution of the three similar PULs; *B. theta* BT4240-50, *B. faecis* KCYDRAFT_01771-81 and *B. caccae* BACCAC_01835-46, was found across 73% of the human gut metagenomes analysed, suggesting that the encoded proteins play an important role in microbial survival in the gut (Supplemental Fig. 12) as shown for other common PULs targeting other substrates[24,25].

Operon 1 of PUL BT4240-50 (BT4240-43) is also found among different *Bacteroides* species including those without homologues of the operon 2 BT4244-47 in the direct neighbourhood, or without operons with BT4244-M60L homologues altogether (e.g. *B. ovatus*) (Supplemental Fig. 13). This suggests that both operons evolved independently from each other and are currently only known to be combined into one PUL in *B. theta*, *B. faecis* and *B. caccae*. For additional PULs encoding BT4244-M60L homologues from *B. theta* (two PULs) and other *Bacteroides* spp., alternative predicted CAZyme associations where identified, suggesting targeting of different glycoproteins/glycan structures to those described for PUL BT4240-50 (Supplemental Fig. 13).

### Association of PUL BT4240-50 and *Bacteroides spp.* with inflammatory bowel disease

The role of the PUL BT4240-50 in mucin breakdown and the link between excess mucin grazing by the microbiota and disease[15,70] led us to analyse the abundance of the PUL in inflammatory bowel disease (IBD) patients. A significant under-representation of the three *Bacteroides* PULs encoding M60L in combination with GH109, GH2, and GalNAc kinase was observed among 39 IBD patients compared to healthy controls (*p*-value 0.0065) (Supplemental Fig. 12). This finding was primarily driven by the 27 ulcerative colitis (UC) patients (*p*-value 0.0349), while no significant difference was noted in the 12 Crohn's disease (CD) patients, though a similar trend was observed (*p*-value 0.0947). These PULs are likely core genes for these three *Bacteroides* species (as shown for *B. theta* across several genomes), hence their presence may reflect the biomass of these bacteria in metagenomic datasets. While these findings are intriguing and suggest that the underrepresentation of the three *Bacteroides* species encoding the PUL BT4240-50 and its two related PULs, may have an association in some way with UC, interpretation is limited by small sample size, dataset noise, and the inability to confirm active live cell counts. Further analysis of global transcript abundance (reflecting more closely active live bacteria) for *B. theta* and *B. caccae* using the IBD-TaMMA metatranscriptomics metanalysis tool[71] (among a total of 18 *Bacteroides* species considered in IBD-TaMMA), revealed some significant distinct patterns between IBD types and gut samples sites when compared to healthy controls (Table S5). Among UC patients, faecal transcript levels showed no significant differences compared to controls for either *B. theta* or *B. caccae*, though there was a trend of overrepresentation for the latter. For CD patients, *B. theta* transcripts were significantly overrepresented in their faecal samples (*p*-value 0.035, FDR 0.067). Other *Bacteroides* species, like *B. fragilis*, were more significantly overrepresented (*P*-value 8.92E-29, FDR 2.13E-27), while others, such as *B. coprosuis*, were significantly underrepresented across both patient groups (Table S5). Colon-derived metatranscriptomic datasets showed only marginal differences in *Bacteroides* species abundance between IBD patients and controls, with no significant changes for *B. theta* or *B. caccae*. In contrast, ileum-derived datasets revealed striking differences; all 18 *Bacteroides* species transcripts were significantly overrepresented in UC patients, with *B. theta* being the most prominent (log2FC 8.76, *p*-value 3.94E-247). Conversely, 10 species, including *B. theta* and *B. caccae*, were significantly underrepresented in the ileum of CD patients, with *B. caccae* being most affected (log2FC -3.64, *p*-value 8.62E-82).

A comparison between ileum and faecal samples revealed that all 18 *Bacteroides* species considered in the IBD-TaMMA tool were massively overrepresented in the transcriptome of faecal (and by inference in the colon too) samples compared to ileum samples (Log2FC range 11–15, with all *p*-values equal to zero). This stark prevalence difference highlights how *Bacteroides* species typically dominate downstream colonic gut regions[15,70–72].

## Discussion

Despite the importance of mucolysis to human health, our understanding of the mechanisms of mucin breakdown by the microbiota is still very limited[14,17,28–31,73–77]. Previously we and others have characterised the founding member BT4244-M60L of a novel family of proteases termed M60-like capable of degrading mucins[32]. BT4244-M60L's association with PUL BT4240-50 suggests that the enzyme acts in concert with the PUL encoded CAZymes, therefore it was expected that the characterisation of the PUL BT4240-50 would not only shed new light into the mechanism of a mucin inducible PUL but also contextualise the proteolytic activity BT4244-M60L. Our data from the biochemical characterisation of various PUL BT4240-50 components suggest that it is capable of metabolising mucin glycopeptides containing the core 1 sugar (T-antigen), as well as the core GalNAc (Tn antigen). Significantly, our combined cell localisation, biochemical, structural, and genetic data support a model where extracellular cleavage of the mucin peptide backbone is coordinated with surface binding and uptake of T-/Tn-containing glycopeptides and their subsequent deglycosylation in the periplasm. The pathway is analogous to the classical PUL paradigm whereby a large glycan is endolytically cleaved by an extracellular enzyme to facilitate import by the OM SusC/D apparatus into the periplasmic space for further processing. However, while endo-glycanases are generally widely involved in this activity, the use of endopeptidases in this case is consistent with the nature of the target substrate, in this case a glycoprotein reflecting the versatility and adaptability of *B. theta* to different nutrient conditions. It is also worth noting that surface exposed endo-acting *O*-glycanases in family GH16 have recently[14] been shown to also be involved in mucin processing at the cell surface, but these enzyme target the *O*-glycan side chains to likely leave short core structures on the peptide backbone, which would then be substrates for glycopeptidases like BT4244-M60L, implying these different classes of enzyme act in concert to fully degrade mucins (Fig. 9). While the substrate for SusC/D transporters appears more commonly to be oligosaccharides, import of peptides by SusC/D systems has previously been shown with RagA/B from *Porphyromonas gingivalis* as well as in the structure of a SusC/D (BT2263-64) of unknown function from *B. theta*, supporting the model of glycopeptides as the target for the BT4246-BT4247 SusC/D as proposed here[44,78]. The structure of BT4246-SusD is notable as this is one of the few SusD structures solved with ligand bound, and only the second SusD that has been structurally characterised with a mucin-derived ligand. The only other SusD structure from a mucin activated PUL is BT1043 (21% id to BT4246) which was solved bound to LacNAc (PDB 3EHN)[55]. While the mucin structures targeted by the BT1043 PUL is not known, it is interesting to note that BT1043 recognises the reducing end sugar (GlcNAc) in the disaccharide, opposite to the non-reducing Gal targeting of BT4246, suggesting BT1043 recognises free mucin oligosaccharides rather than glycopeptides like BT4246.

Secreted proteases targeting non-glycosylated terminal domains of mucins are generally recognised as key virulence factors for a variety of mucosal pathogens, allowing disruption of the cross-linked mucin chain network to facilitate access to the epithelium[5,10,50]. Restricted targeting of the terminal domains is thought to be due to the heavy *O*-glycosylation of the central PTS rich region which protects it from proteolysis. By contrast, BT4244-M60L and other M60-like glycopeptidases encoded by the microbiota are able to cleave the glycosylated PTS backbone of the intestinal mucins and may provide an explanation for the report that very little intact mucin is found in faeces[5], a fact that is at odds with an exclusively extracellular glycosidase model of mucin breakdown, which would result in the peptide

backbone remaining largely intact. The capacity to fully process mucins substrates by members of the microbiota also ensures the recycling of an important fraction of the large biomass of secreted mucins, a process which contributes to gut-microbiota homoeostasis both in terms of energy and as a carbon source for the microbiota and enterocytes through short chain fatty acid production[79]. We also demonstrate that BT4244-M60L can cleave the O-glycosylated peptide linker of IgA$_1$, providing potential new insights into the complex mechanisms of glycoprotein processing by the microbiota. Notably, BT4244-M60L preferentially targets PTS with Tn-antigen and core 1 (T antigen) decorations in IgA$_1$ and does not efficiently cleave sialylated or unglycosylated structures, consistent with the recent data from several other studies looking at BT4244-M60L specificity[34–36,33]. IgA$_1$ is also a prominent mucosal glycoprotein and has been shown to be targeted by proteases from several pathogenic bacterial species. Degradation of IgA$_1$ is thought to represent an immune evasion mechanism in these bacteria. Interestingly, BT4244-M60L cleaves at the same site as a *Prevotella melaninogenica* protease and the pathogenic bacteria *Haemophilus aegyptius*, HI-1 and *Haemophilus influenzae*[37,80,81]. Whether this PUL enables *B. theta* to extract glycans from IgA$_1$ or evade immune action is not known and the functional significance of its IgA$_1$ protease activity is yet to be established, although *B. theta* cells grown on PGMIII were able to partially degrade the immunoglobulin, suggesting processing of IgA1 by *B. theta* may occur in the gut. The successful exploitation of IgA$_1$ in understanding BT4244-M60L activity however, demonstrates that it could serve as a useful alternative or proxy substrate in place of synthetic glycopeptides for screening and analysing glycoproteases[33,82], especially those from the M60-like protease family where most members remain uncharacterised to date.

Genetic studies of PUL BT4240-50 demonstrated a critical role played by the PUL in human and animal host-derived glycoprotein metabolism, particularly in providing competitive access to the prominent mucin and CS sugar GalNAc. This is supported by in vitro competition or fitness data using PGMIII as the sole carbon source, inability of the deletion mutant ΔPULBT4240-50 and ΔBT4240 to grow on GalNAc as the sole carbon source and complementation experiments showing that BT4240-kinase enzyme is capable of significantly restoring the growth defects caused by the deletion of PUL BT4240-50. This also implies that the majority of the other PUL components exhibit redundant activities during growth on PGMIII in vitro, which is not surprising given the significant expansion of mucin PULs and GH families in the *B. theta* genome and data showing that BT3169-GH31, which has previously been implicated in mucin breakdown, was also capable of extracting core GalNAc from mucins[83]. It was however very interesting that despite the large number of mucin PULs in *B. theta*, a single enzyme BT4240-kinase was responsible for the metabolism of such a critical and core component of mucins as GalNAc. Indeed, all mucin O-glycans contain a GalNAc anchor, implying that BT4240-kinase is likely a key player in competitive colonisation by *B. theta* in vivo. The involvement of BT4240-kinase in global GalNAc metabolism was further supported by the striking observation that the ΔBT4240 mutant strain showed ~50% reduction in growth on CS, directly reflecting the proportion of GalNAc in the polysaccharide.

While the characterisation of PUL BT4240-50 provides new insight into the mechanism of mucin breakdown by the *B. theta*, it is only one piece of a rather large and complex puzzle, with many PULs containing multiple putative CAZymes, sulfatases and proteases activated during growth on mucin O-glycans[8,17,28]. These findings highlight both the complexity and heterogeneity of these host derived substrates and the fact that unlike most glycans, the breakdown of mucins appears to require the cooperative action of multiple PULs, each likely targeting a discrete structure. Indeed, for PUL BT4240-50 to access its proposed target the short core 1 and Tn structures in mucins, it would

first require significant 'debranching' of the heterogeneous glycan side chains of mucin (Fig. 1 and Fig. 9) by enzymes from other mucin-targeting PULs in *B. theta* and likely other members of the mucolytic microbiota. Similarly, it is possible that components of the PUL BT4240-50 are utilised by *B. theta* to access structures other than core 1. For instance, as the GalNAc-Ser/Thr structure is conserved in all mucin cores[40] (Fig. 1), the BT4243-GH109 (and BT3169-GH31) could also be used by *B. theta* to release GalNAc from other core structures once the capping structures had been removed by other enzymes from mucin targeting PULs. BT4240-kinase however, appears to represent the only GalNAc kinase in *B. theta*, suggesting it is essential for the metabolism of this monosaccharide derived from various sources such as other core structures, terminal epitopes or even other types of glycan such as CS[22,65,84] and bacterial lipopolysaccharides[85,86]. These observations could explain the higher basal level expression of operon 1 encoding BT4240-43 (the kinase and GHs) vs operon 2 (BT4244-47) which is more tightly controlled and appears to encode more mucin-specific activities (Fig. 2A)[17].

Regarding the types of mucin targeted by PUL BT4240-50, it is worth noting that core 1 structures though present, are relatively rare in human MUC2, the major colonic mucin, which mainly comprises core 3, and to a lesser extent core 2 and 4, suggesting other *B. theta* PULs play a more significant role in the degradation of colonic MUC2[57,87]. Interestingly, among the *B. theta* PULs known to be induced by mucins there are an additional three candidate surface proteases which could cooperate with BT4244-M60L (including a homologue of BT4244-M60L, BT3015; Supplemental Fig. 13 and Table S4) to degrade different parts of the MUC2 polypeptide backbone and/or other mucin types. Notably, core 1 is common in secreted mucins from the upper digestive tract (e.g. MUC5B from saliva and MUC5AC from the stomach respectively)[88,89] and thus PUL BT4240-50 could play an important role in the breakdown of these non-colonic mucins as they converge in the colon (i.e. after travelling down the digestive tract), somewhat mirroring exposure of *B. theta* to dietary glycans. Similarly IgA$_1$, with mucin-like PTS repeats decorated with core 1[49] which was shown here to be a substrate for BT4244-M60L, is by far the more abundant isotype of secretory IgA in the upper part of the digestive (and respiratory) tracts (60–95%), with the colon being the only site of the digestive tract where the non-O-glycosylated IgA$_2$ is more abundant than IgA$_1$ (~60:40)[90].

Strikingly, 73% of scanned human gut metagenomes possess at least one of the three identified BT4240-50-like PULs (Supplemental fig. 12), supporting the general importance of this apparatus for gut survival. Furthermore, the absolute conservation of PUL BT4240-50 across all *B. theta* genome sequences from isolates from both human and animal guts, and human faecal metagenomes datasets, is also consistent with the locus playing a central role in the biology of *B. theta*, contrasting with PULs with a more ephemeral status such as those targeting some diet-derived plant glycans[69] or a fungal glycan targeting PUL[24]. The functional importance of PUL BT4240-50 in mucin breakdown was recently further highlighted in a mouse model investigating the influence of diet on the evolution of the *B. theta* genome[91]. A Western-style diet (WD), characterised by high fat, low fibre, and an abundance of the simple sugars, is known to promote the utilisation of host mucins as a nutrient for the microbiota[15,17]. Within three months of WD exposure, a significantly increased mutation frequency was observed in the BT4246-7 SusC/D genes, pointing to a central/master role of PUL BT4240-50 in mucin consumption, compared to the other 15–17 potential mucin-targeting PULs encoded by *B. theta*, where the SusC/D genes were not affected by the low fibre diet. Notably, these WD specific mutations enhanced the bacterium's ability to colonise the mouse gut by improving its efficiency in utilising mucin-derived glycans, compensating for the lack of dietary fibres in the WD[91]. Intriguingly, most of these mutations are in BT4247 SusC at the surface 'lip' of the barrel (Supplemental Fig. 14), suggesting this

location plays a key role in modulating the efficiency of mucin glyco-peptide uptake.

The survey of human faecal metagenomes highlighted the high abundance of the PUL BT4240-50 and the two related PULs from *B. caccae* and *B. faecis* (>70%) across human populations, suggesting an important role in the microbial ecosystem of these three species in the human colon – e.g. contributing to initial colonisation and ecosystem stability as they can access an abundant and reliable source of host derived nutrients. A significant underrepresentation of these three PULs in faecal metagenomes from UC patients is consistent with a positive regulatory role of these bacteria in gut homoeostasis[92]. We complemented these analyses with analysis of comparative meta-transcriptomics data at different sites of the gut to consider patterns associated with active cells, which likely better reflect bacterial functionality in the gut. The significant overrepresentation of *B. theta* and *B. caccae* (and all 16 additional *Bacteroides* species in IBD-TaMMA) in metatranscriptomics datasets from the ileum of UC patients was unexpected and suggests potential dysfunction in the ileum's defence mechanisms that typically control strict anaerobes[93]. The ability of *B. theta* and *B. caccae* to efficiently degrade mucins, facilitated by the PULs described here, points to the possibility of excessive mucin grazing in the thinner and looser mucus layer of the ileum in UC patients. This excessive degradation may alter the immunological environment of the ileum, potentially contributing to the inflammatory cascade characteristic of UC. Such changes in the ileal immune status could represent an important factor exacerbating the damaging inflammation observed in the terminal colon of UC patients[92]. In contrast, *B. theta* and *B. caccae* and 8 additional *Bacteroides* species, were all significantly underrepresented in the ileum of CD patients. This suggest different contributions by these species in gut homoeostasis dysregulation among CD patients compared to UC patients[91–93].

To conclude, we have functionally characterised one of several uncharacterised mucin inducible PULs BT4240-50 from the genome of the prominent human gut microbe *B. theta*, revealing key mechanistic insights into how it coordinates the processing of mucin and other glycoprotein structures. Critically, we have demonstrated that PUL BT4240-50 is essential for in vitro competitive growth on mucin glycans by *B. theta* and encodes a key kinase enzyme which makes a major contribution to mucin and GalNAc metabolism by phosphorylating the latter and enabling its utilisation as an energy or carbon source or both, promoting competitive growth. These data advance our knowledge of the vital metabolic processes that govern host-microbiota interactions at mucosal surfaces and highlights GalNAc as a key metabolite mediating microbiota-gut interactions, which could represent a significant factor in modulating some inflammatory conditions such as IBD.

## Methods

### Sources of carbohydrates
PGMII and PGMIII, BSM, LacNAc, Lacto-N-biose, core 1 disaccharide (galacto-N-biose), α-Galactobiose, monosaccharides and PNP-linked monosaccharides were from Sigma. T-antigen (Galβ1,3GalNAcα1-Ser) and Tn-antigen (GalNAcα1-Ser) were purchased from Dextra Laboratories (Reading, UK) and PNP-GlcNAcβ1,3GalNAc (PNP-core 3) from Santa Cruz Biotechnology (Dallas, Texas, USA). β1,3-galactobiose was a generous gift from Dr Jose Munoz (University of Northumbria, UK).

### Cloning, expression and purification of recombinant proteins
The genes encoding the mature forms (i.e. lacking their predicted signal sequences) of the BT4240-4250 proteins characterised in this study were amplified from *B. theta* VPI-5482 genomic DNA using the primers shown in Table S6. Note that the database sequence for BT4241-GH2 has the wrong N-terminal methionine annotated as no signal peptide is predicted, but when the methionine (21 residues upstream) is used i.e., sequence starting MKYR, it has a strongly

predicted Type I signal peptide (predictions used LipoP, https://services.healthtech.dtu.dk/services/LipoP-1.0/ and SignalP 6.0, https://services.healthtech.dtu.dk/services/SignalP-6.0/). The PCR products were cloned into pET28 (Novagen) or pRSETA (ThermoFisher) expression vectors and sequenced to ensure the fidelity of the cloning process. Recombinant plasmids were transformed into BL21 (DE3) or BL21 (DE3) Tuner cells (Novagen) and were incubated in 1 L LB in 2 L conical flasks at 37 °C, 180 rpm until $OD_{600}$ - 0.6. The cultures were then cooled to 16 °C before addition of 1 mM isopropyl-β-D-thiogalactoside (IPTG) to induce recombinant gene expression and incubation at 16 °C, 180 rpm overnight. Cells were harvested by centrifugation (3000 $g$, 4 °C), resuspended in Talon buffer (20 mM Tris-HCl, pH 8.0, 100 mM NaCl) and lysed by sonication. Recombinant His-tagged protein was purified from cell free extracts in a single step using immobilized affinity chromatography (IMAC) with Talon resin (Clon-Tech) and dialysed overnight into 20 mM Tris-HCl, pH 8.0. The amount of purified protein was quantified using the predicted molar extinction coefficient as determined by the ProtParam tool (Expasy server). Proteins were used fresh or stored in aliquots at -20 °C until required.

### Growth and genetic manipulation of *B. theta*
*B. theta* was routinely cultured under anaerobic conditions in 5 ml of TYG (tryptone-yeast extract-glucose medium) or minimal medium (MM) containing 0.5–1% (w/v) of an appropriate carbon source plus 1.2 mg/ml porcine haematin (Sigma) in glass test tubes as described previously[22]. Growth of cultures was monitored using a Biochrom WPA cell density meter (Cambridge, UK).

*B. theta* knockout strains were created using counter-selectable allelic exchange as described previously[94]. Signature-tagged strains of Wt and ΔBT4240_50 *B. theta* mutant used in competition experiments were created and selected by integration of unique 24 bp tag sequences into one of two *att* sites in the genome of *B. theta* following conjugation with an S17-1 *λ pir E.coli* strain harbouring an NBU-2 based vector (pNBU2-tetQb) as described previously[17]. Selection of integrants was done on BHI-agar containing gentamycin (200 mg/ml) and tetracycline (1 mg/ml). Genomic DNA was extracted from single colonies after culturing in TYG using the GenElute™ Bacterial Genomic DNA kit (Sigma) and screened for tag insertions by PCR using primers flanking various *att* sites. A list of all primers used in various genetic manipulations is provided in Table S6.

### In vitro competition of Wt and ΔBT4240-50 *B. theta* strains on mucin
Competition experiments were carried out to evaluate the contribution of the BT4240-50 locus to *B. theta* fitness on PGMIII and glucose. Approximately equal amounts of the signature-tagged Wt and ΔBT4240-50 deletion mutant initially grown in TYG were mixed and 100 μl used to inoculate minimal medium containing 1% glucose (MM-Glc). After overnight growth, 100 μl of culture was used to inoculate fresh MM-Glc media every day for 5 days before switching to MM-PGMIII (1% w/v) for a further 5 days. Genomic DNA was extracted from 2 ml of overnight culture from each time point using the GenElute™ Bacterial Genomic DNA kit (Sigma) and 10 ng analysed by qPCR using a Light Cycler 480 real-time PCR system (Roche) to quantify the proportion of each tagged strain.

### Cellular localisation studies
The expression and cellular localisation of BT4244-M60L was determined using rabbit polyclonal antibodies (Eurogentec) generated against a purified recombinant version of the M60-like domain of BT4244-M60L. *B. theta* cells grown on MM-containing PGMII to mid exponential phase ($OD_{600}$ - 0.4) where fixed in an equal volume of 9% formalin in PBS, pH 7.4, by rocking gently for 90 min at 25 °C. Cells were washed twice with PBS then pelleted by centrifugation for 5 min at 2400 $g$ and re-suspended in 1 ml of PBS. This washing process was

repeated twice before incubation with blocking solution (1 ml of 2% normal goat serum, 0.02% NaN₃ in PBS) overnight at 4 °C. Cells were then centrifuged again at $17000\,g$ for 1 min and blocking solution discarded. For labelling, cells were incubated with 0.5 ml of a 1/1000 dilution (in blocking solution) of rabbit polyclonal anti-BT4244-M60L antibodies for 2 h at 25 °C. Cells were washed again as above by centrifugation at $17,000\,g$ for 1 min. Detection of bound antibodies was carried out using a 1/200 dilution of Alexa-Fluor® 594 conjugated goat anti-Rabbit IgG secondary antibodies (Molecular Probes) by incubating for 1 h at 25 °C in the dark. Cells were then pelleted and washed with PBS as above. To 100 μl of cells, one drop (~50 μl) of ProLong Gold anti-fade reagent (Life Technologies) was added and labelled bacterial cells were mounted onto glass slides for imaging. Phase contrast and fluorescence images were captured using an Andor iXonEM+ 885 EMCCD camera coupled to a Nikon Ti-E microscope.

FLAG® tagging of the C-terminus of BT4240-4250 encoded proteins was done by incorporating the FLAG® peptide (DYKDDDDK) DNA sequence into the genome of *B. theta* using the primers listed in Table S6. Strains containing FLAG®-tagged proteins were grown overnight in 10 ml cultures MM-PGMIII (1% w/v), harvested by centrifugation ($2400\,g$, 10 min, 4 °C), washed in PBS buffer and disrupted by sonication for 2 min on ice. This was followed by the separation of membrane and soluble fractions using ultracentrifugation as described previously[95,21]. Fractions were analysed by Western blotting using a 1/10,000 dilution of primary rabbit anti-FLAG® antibodies (F7425, Sigma) followed by a 1/5000 dilution of secondary donkey anti-rabbit IgG conjugated to HRP (Santa Cruz biotechnology, USA).

### Glycoside hydrolase and sugar kinase assays

**TLC and HPAEC-PAD.** TLC analysis of sugars was performed using aluminium backed silica TLC plates (Sigma)[66]. Reactions (3–6 μl of each) were spotted on to plates and samples resolved in butanol/acetic acid/water buffer (2:1:1) in glass tanks. Plates were then dried with a hair dryer and dipped in either orcinol/sulfuric acid (0.5% (w/v) orcinol in 10% sulfuric acid) or diphenylamine–aniline–phosphoric acid (DPA, used to improve detection of sialic acid and other N-acetylated sugars which are generally not very visible with orcinol[96].) developers before being oven dried at ~70 °C for up to 10 min to visualise sugars[96].) developers before being oven dried at ~70 °C for up to 10 min to visualise sugars. Reactions (50 μl total) were set up in 1.5 ml microcentrifuge tubes containing enzyme (BT4241 or BT4243 at 0.3–1.2 μM) and substrate (5-10 mM disaccharides or 4 mg/ml mucins) in 20 mM Tris-HCl, pH 7.5 and incubated at 37 °C for 1 h. The sialidase used for pretreatment of BSM was from *Arthrobacter ureafaciens* (Calbiochem) or *B. theta* (BT0455)[73] and the fucosidase used for pretreatment of PGMII was from *Bifidobacterium bifidum*[97]. For HPAEC-PAD, samples were analysed using a Dionex DX-500 ion chromatography system (Dionex) fitted with a CarboPac™ PA1 anion-exchange column (ThermoFisher). The run consisted of two elution phases, an isocratic elution phase with 20 mM NaOH followed by a gradient phase with increasing concentration of 500 mM NaOAc (0–100%) in 20 mM NaOH. The flow rate was set to 0.4 ml/min and eluates were monitored by pulsed amperometric detection (PAD) with a fitted electrochemical detector. HPAEC-PAD data were analysed using Chromeleon™ chromatography software (version 6.8, Dionex).

**Enzyme kinetics.** The kinetic parameters of the BT4241-GH2 β-galactosidase against disaccharide substrates was determined by the continuous monitoring of Gal release using a D-Gal detection kit (Reference code: K-ARGA; Megazyme International) at 37 °C in 20 mM MOPS buffer, pH 7.0. For core 1, LNB and β1,3-galactobiose disaccharides reactions contained 0.3 μM of BT4241-GH2 and a range of substrate concentrations from 0.05-2.0 mM. For LacNAc, 3 μM of enzyme was used with substrate concentrations ranging from 0.4 to 6.0 mM. To screen enzymes against paranitrophenol (PNP)-linked monosaccharides, 1 μM enzyme (final) was added to 0.5 mM substrate and PNP release determined at 37 °C in 20 mM MOPS buffer, pH 7.0, by continuously monitoring the increase in $A_{400}$. For BT4243-GH109 kinetics, 0.4 or 0.7 μM enzyme (final) was added to substrate (PNP-α-GalNAc or PNP-β-GalNAc, respectively) at a range of concentrations from 0.03-0.5 mM. BT4240 kinase activity was determined using a coupled enzyme ATPase assay as described previously[98]. BT4240-kinase (between 0.1 and 0.5 μM final) was added to a range of monosaccharides or amino acids (between 0.2–50 mM) at 37 °C in 20 mM MOPS buffer, pH 7.0 and the reaction rate monitored by the decrease in $A_{340}$ from the conversion of NADH to NAD. Absorbances were converted to mols of product by using the extinction coefficient of NADH (6220 M⁻¹ cm⁻¹ at $A_{340}$) or PNP (16,800 M⁻¹cm⁻¹ for Talon pH 8.0 and 9000 M⁻¹ cm⁻¹ for MOPS, pH 7.0 at $A_{400}$) where applicable. Graphs of initial rate versus substrate concentration were generated and fit to the Michaelis–Menten equation using GraphPad Prism v6.0 to determine enzyme kinetic parameters. BT4240-kinase activity was determined using a coupled enzyme ATPase assay as described previously[98]. BT4240-kinase (between 0.1-0.5 μM final) was added to a range of monosaccharides or amino acids (between 0.2 and 50 mM) at 37 °C in 20 mM MOPS buffer, pH 7.0 and the reaction rate monitored by the decrease in $A_{340}$ from the conversion of NADH to NAD. Graphs of initial rate versus substrate concentration were generated and fit to the Michaelis–Menten equation using GraphPad Prism v6.0 to determine enzyme kinetic parameters.

### IgA cleavage assay

Human myeloma IgA₁ or IgA₂ isoforms (Calbiochem), at 0.33 mg/ml were separately incubated with 1 μM of recombinant BT4244-M60L peptidase in Talon buffer pH 8.0 at 37 °C in a water bath for 16 h. Samples were then analysed by SDS-PAGE followed by Western blotting. For Western blotting, PVDF membranes containing blotted proteins were washed in excess PBS-Tween 20 for 1 h at room temperature before application of a 1:2000 dilution of primary mouse Anti-Human IgA₁ antibodies conjugated to biotin (SouthernBiotech). Blots were washed 10 min 3 x in PBS followed by treatment with a 1:2000 dilution of ExtrAvidin®–Peroxidase conjugate (Sigma) in washing solution for 1 h. A final washing step was performed as above before chemiluminescence detection with luminol/enhancer solutions from the Biorad Immun-Star™ Western C™ Chemiluminescence kit (Bio-rad) and Luminol and enhancer solutions were mixed in equal proportions (1 ml each), spread on washed membranes and chemiluminescent signals detected using a ChemiDoc XRS system (Bio-rad). To determine the effect of deglycosylation of IgA₁ on BT4244-M60L peptidase activity, IgA₁ was pre-treated with a sialidase from *Clostridium perfringens* (Sigma), BT4241-GH2 β-galactosidase (0.5 μM) and BT4243-GH109 α-N-acetylgalactosaminidase (1 μM) enzymes in different combinations overnight in a final volume of 15 μl in 20 mM Tris-HCl, pH 8.0, 100 mM NaCl, before incubation with BT4244-M60L glycopeptidase (0.7 μM) for 16 h. IgA deglycosylation was monitored using a biotin–conjugated *Helix aspersa* agglutinin (HAA) (Sigma, cat. L8764) that binds specifically to GalNAc. This was performed on separate blots containing replicate samples and using a 1:1000 dilution of the lectin. For IgA₁ degradation by *B. thetatiotaomicron*, cells were cultured in minimal medium containing 1% w/v PGMIII or Glc with and without IgA₁ at 37 °C overnight in an anaerobic chamber. Next day, cultures were centrifuged, and supernatants collected and resolved on 10% SDS-PAGE gels and stained with Coomassie blue reagent.

### Isothermal titration calorimetry (ITC)

ITC was performed using a MicroCal VP-ITC machine. Titrations (25 × 10 μl injections) were carried out in 20 mM Tris-HCl buffer, pH 7.5 at 25 °C. The reaction cell contained protein at 50–100 μM, while the

syringe contained ligand (between 2–50 mM). Integrated heats, minus control heats of dilution, were fit to a single site binding model using Origin v7.0 to derive $K_a$.

## Cloning and expression of BT4246 in *E. coli* for crystallisation

The gene fragment corresponding to BT4246 lacking its endogenous signal peptide (residues 25–642) was amplified from *B. theta* genomic DNA using the primers shown in Table S6. The gene product was ligated into a modified version of pET-28a (EMD Biosciences) containing a tobacco etch virus (TEV) protease recognition site. The pET28-BT4246 plasmid was transformed into Rosetta (DE3) pLysS cells (EMD Biosciences). Transformed cells were grown at 37 °C for 20 h, and then the plates were scraped to inoculate culture media for protein expression. For native protein expression, the cells were grown in 1 L of LB containing 50 mg/ml Kanamycin and 20 mg/ml Chloramphenicol (in 2 L baffled flasks) at 37 °C until they reached an $OD_{600}$ of ~0.6. The culture was cooled down to 22 °C and protein expression induced by adding 0.5 mM IPTG. Induced cells were cultured overnight (~16 h). Next day, cells were harvested by centrifugation at 6000 $g$, and the cell pellets were stored at -80 °C until protein purification. SelenoMet-substituted protein was produced using a premade SelenoMet expression medium (Molecular Dimensions). Cells were grown for 16 h in 100 ml Minimal Media starter cultures containing 40 μg/ml methionine. Cells were then pelleted at 6000 $g$ and washed three time with sterile $H_2O$ then transferred to 1 × 2 litre baffled flasks containing the SelenoMet expression media supplemented with 40 μg/ml of SelenoMet. The cells were grown at 37 °C until they reached an $OD_{600}$ ~ 0.6, then the temperature was lowered to 22 °C and the cells were grown for an additional 16 h. Cultures were pelleted via centrifugation and stored at -80 °C prior to purification.

## Purification of native and SelenoMet-substituted BT4246

Native and SelenoMet-substituted BT4246 proteins were purified using a 5 ml Hi-Trap metal affinity cartridge (GE Healthcare) according to the manufacturer's instructions. The cell lysate was applied to the column in His Buffer (25 mM $NaH_2PO_4$, 500 mM NaCl, 20 mM imidazole, 1 mM TCEP pH 7.5) and proteins were eluted with an imidazole gradient (20–300 mM). The His-tag was removed by incubation with recombinant TEV (1:100 molar ratio of TEV to protein) at room temperature for 3 h, then for 16 h at 4 °C while dialyzing against His Buffer. The cleaved protein was re-purified on the 5 ml Ni column to remove undigested target protein, the cleaved His-tag and His-tagged rTEV. Purified proteins were dialysed against 20 mM HEPES, 100 mM NaCl pH 7.0 prior to Crystallisation, and concentrated using Vivaspin 15 (10,000 MWCO) centrifugal concentrators (Vivaproducts, Inc.).

## Crystallisation and data collection

Crystallisation conditions were screened via the hanging drop method of vapour diffusion in 96-well plates and using Hampton Screen kits (Hampton Research). Native crystals were obtained at 4 °C via hanging drop vapour diffusion from protein concentrated to 8.8 mg/ml and mixed 1:1 with the well solution containing 22–24% PEG 3350 and 150 mM CsCl. To obtain a ligand-bound structure, we soaked these native crystals in a 4% solution of oligosaccharides derived from PGM, prepared by alkaline borohydride treatment as previously described[17]. Briefly, *O*-glycans were prepared from PGMIII, solubilised in 100 mM Tris, pH 7.4 (2.5% w/v), and autoclaved, followed by Proteinase K treatment at 65 °C. Insoluble material was removed by centrifugation, followed by NaOH and NaBH4 treatment of the soluble fraction to release *O*-glycans via β-elimination. After neutralisation and centrifugation, the *O*-glycans were subjected to two rounds of anion exchange chromatography (DEAE-Sepharose, Sigma). *O*-glycan oligosaccharides used for crystallisation were from the flow-through fraction (50 mM Tris, pH 7.4) that represents neutral sugars. SelenoMet-substituted proteins crystals were obtained at 20 °C as hanging drop

experiments using 22.56 mg/ml protein against a well solution of 22–24% PEG 3350 and 200 mM KCl, using grains of $SiO_2$ in the hanging drop to drive nucleation. These crystals were then micro-seeded onto a hanging drop containing 9.92 mg/ml of SelenoMet substituted protein and $SiO_2$ against a well solution of 23% PEG 3350 and 200 mM KCl. Crystals that nucleated half way off of the $SiO_2$ were then transferred to a hanging drop containing 17.28 mg/ml of SelenoMet substituted protein, against a well solution of 20% PEG 3350 and 200 mM KCl. All crystals were serially transferred into a cryoprotectant composed of 80% crystallisation media, 20% ethylene glycol and flash-frozen in liquid nitrogen prior to data collection.

X-ray diffraction data were collected at the Life Sciences Collaborative Access Team (LS-CAT) at the Advanced Photon Source at Argonne National Labs, Argonne, IL. X-ray data were processed with HKL2000 and scaled with SCALEPACK[99]. The structure was determined from the SAD data using the AutoSol subroutine within the Phenix software package[100,101]. The SelenoMet-substituted model of BT4246 was then utilised for molecular replacement in Phaser[102] against the native X-ray data sets. The final models for all structures were derived from iterative cycles of manual model building in Coot[103] and refinement in Phenix. Data collection and refinement statistics are reported in Table S2. A portion of the electron density for each of the BT4246 SusD-like structures solved in this study is shown in Supplemental Fig. 15.

## Bioinformatic analyses

Metagenomic analyses of the occurrence of the BT4240-50 PUL and selected PULs for comparison from *B. theta* and other *Bacteroides* spp. in human gut metagenome data was performed as described below and in two related publications[24,25]. The BT4240-50 PUL corresponds to PUL 78[17] and is also described in PULDB[104]. The related PUL from *B. faecis* MAJ27: KCYDRAFT_01771-81 (predicted PUL 70 *B. faecis* strain 23150, which is included in PULDB) and *B. caccae* ATCC43185: BACCAC_1835-46 (predicted PUL 34 in PULDB) were also considered. The PUL DNA sequences were used as query in BLAST searches (word size of 11) and hits were recorded when they passed the following cutoff criteria: $E$ values < 1E-20, and nucleotide identities >90% over a length >100 bp. A PUL was considered present in a given sample when at least 2 distinct BLAST hits were recorded over the length of the query sequence. The details for the query sequences are as follows: *B. theta* VPI-5482, PUL *BT_4240-50*, gb|AE015928.1|, positions 5582950-5601300, 18,350 bp. *Bacteroides faecis* MAJ27, PUL *KCY_RSO108600-KCY_RSO108650* (*BT4240-50*-like), ref|NZ_AGDG01000020.1|, positions 51630-69946, 18,316 bp. *Bacteroides caccae* ATCC 43185, PUL *BACCAC_01835-46* (*BT4240-50*-like), B_caccae-MSIQ_Cont1328, gb|AAVM02000003.1|, positions 306812-328036, 21,224 bp. *Bacteroides caccae* ATCC 43185, inulin PUL *BACCAC_02727-31*, gb|AAVM02000006.1|, positions 41899-59372, 17,473 bp. *Bacteroides plebeius* DSM 17135, porphyran PUL *BACPLE_01672-708*, gb|ABQC02000019.1|, positions 132998-78523, 54,476 bp. *B. theta* VPI-5482, α-mannan PUL1 *BT_2620-32*, gb|AE015928.1|, positions 3262908-3277966, 15,059 bp. *B. theta* VPI-5482, α-mannan PUL2 *BT_3773-92*, gb|AE015928.1|, position 4893415-4928241, 34,827 bp. *Bacteroides uniformis* ATCC 8492, xyloglucan PUL1 *BACUNI_00315-26*, gb|AAYH02000032.1|, positions 43453-71885, 28,433 bp.

## Reporting summary

Further information on research design is available in the Nature Portfolio Reporting Summary linked to this article.

# Data availability

All data generated in this study are available in the figures, tables and supplementary information. Crystal structure datasets generated have been deposited in the PDB under accession code PDB; 5CJZ (BT4246). Source data are provided with this paper.

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

## Acknowledgements

We would like to thank Carl Morland for his expert technical assistance, Dr D. Bulmer (Newcastle University) for assistance with microscopy, and Dr A. Zhu (European Molecular Biology Laboratory) for providing the annotation of the 41 reconstructed *B. theta* genomes from human faecal metagenomes. We also thank Nicholas Pudlo and Prof Eric Martens from the University of Michigan Medical School for the generous donation of mucin *O*-glycan oligosaccharides for co-crystallisation with BT4246 and Professors Mike Ferguson and Claire Halpin at the University of Dundee for the kind donation of reagents including phosphorylated sugars and access to vital analytical lab equipment. This work was supported in part by a BBSRC/Innovate UK IB catalyst award to D.N.B. 'Glycoenzymes for Bioindustries' (BB/M029018/1), as well by funds from a pilot/feasibility grant from the University of Michigan Gastrointestinal Peptides Research Center (DK 034933) awarded to N.M.K. and the Host Microbiome Initiative at the University of Michigan Medical School (N.M.K.). D.A.N.'s PhD project was funded by Newcastle University and a Commonwealth Scholarship (CMCS-2010-86). D.A.N.'s lab at the University of Dundee is currently supported by the Royal Society through a University Research Fellowship (URF\R1\221864).

## Author contributions

D.A.N. and C.S. performed enzymology experiments. D.A.N. performed bacterial growth, genetic, cellular localisation and isothermal titration calorimetry experiments N.M.K. and K.J.K. performed structural biology experiments, D.N.B., N.M.K. and D.A.N. produced protein/substrate structural models. R.P.H. and S.N. performed bioinformatic and statistical analyses, D.A.N., D.N.B., N.M.K. and R.P.H. produced figures, D.A.N., D.N.B. and R.P.H. conceptualised the work and wrote the first draft. All authors contributed to editing and revision of the paper.

## Competing interests

The authors declare no competing interests.
