## [Transparent Peer Review file · Nature Communications]

A *Bacteroides thetaiotaomicron* genetic locus encodes activities consistent with mucin O-glycoprotein processing and N-acetylgalactosamine metabolism

Corresponding Author: Dr Didier Ndeh

Version 0:

Reviewer comments:

Reviewer #1

(Remarks to the Author)

Ndeh et al have used a combined genomic, structural, and biochemical approach to characterise the mechanism of action of a key mucin-targeting PUL from *Bacteroides theta*, regarded as a model species for studying the human gut microbiome and a cornerstone of that community. The experimental work has been performed to a high quality and I agree with the conclusions the authors draw from their work. Although the enzyme characterisation results are not surprising, and align well with the predicted functions of the respective proteins, it is interesting to see the data put together and contextualised. Thus, the majority of my comments relate to the presentation of the data, as there are unfortunately a few issues here. There are inconsistencies and lacks of clarity in the text and figures, which undermine the quality of the work. See some specific issues described below, most of which should be straight-forward to address.

-Consider using the more up to date term Bacteroidota instead of Bacteroidetes?

-Line 59, "Central to all mucin O-glycosylated sugars..." Should this be "Central to all mucin O-glycosylated proteins..." ?

-Figure 1, it's a bit unclear which elements belong to panel A.

-You mention in the Introduction and elsewhere that the PUL is likely involved in metabolising GalNac-glycans other than mucin, and the enzymatic evidence for this is clear, but it is not clear to me if it is known whether the whole PUL is highly expressed during growth on other substrates? I apologise if this is mentioned in the text, but if so, it is not clearly presented.

-On a similar note, the Introduction states that one of the two operons in the PUL shows higher basal expression than the other. As far as I can tell, the one with higher basal expression is the one encoding the CAZymes – the Discussion does not seem to come back to this point of there being two separately controlled operons but is interesting and perhaps relevant to my previous comment.

-Line 118, word missing "...encoded proteins are involved IN the acquisition and breakdown..."

-The phrases *in silico*, *in vivo*, and *in vitro* should always be in italics but should never be connected by hyphen (e.g. *in-silico*), please correct throughout.

-There are issues with the ordering/organisation of items in several figures (Fig 1, Fig 2, Fig 3, Fig 8...). Please assume that the reader will look for panels A, B, C, etc in logical reading order, i.e. left-to-right and then top-to-bottom. Figure 3 especially is hard to comprehend, since the panels are not presented in the alphabetical order to which they are referred in the caption, so it feels chaotic to try to read the figure.

-In Fig 2, I think panel F is mislabelled as E? And no panel is labelled F, despite the caption and an in-text reference to Fig 2F.

-Line 163 – where does this lectin come from?

-Line 171 mentions a BACON domain of unknown function – since the BACON name derives from "Bacteroides-associated carbohydrate-binding often N-terminal" I think it is wise to explain the nomenclature a bit more. Why is it called a BACON if it is not expected to bind carbohydrates?

-Line 178 and 179 refer the reader to Suppl Fig S2 to see binding data for the CBM32 and BACON constructs, but the data for the BACON alone are not shown. Also, is "BACON-CBM32" the same as "BT4245-FL"? Please use consistent naming conventions across text, figures, and captions – this is a recurring issue throughout the paper and makes it difficult to keep track of or understand the results.

-Line 185, should this not refer to 'ligand' recognition rather than 'substrate', since it is a CBM being discussed?

-Line 200, word missing, "...partner SusC-like PROTEINS for active..."

-As mentioned above, the layout of Figure 3 is chaotic and hard to follow. Moreover, I can find no reference at all to panels A-

C of the figure in the actual manuscript. Why include those images if they are not relevant to the paper? Please make sure to refer to all figures.

-Figures 3D and 3E are perhaps too small to see the ligand clearly.

-On a similar note to an earlier comment, the sections of figure 4 are not cited in order (Fig 4A line236, Fig 4B line242, Fig 4F line244, Fig 4C line247, Fig 4D line253). Ideally you could rethink the order of panels in the figure so they can be cited in order. Unless I missed it, there is no reference to Fig 4E anywhere in the text.

-Figure 4C is not very compelling if intended to show a different product profile for one vs two enzymes combined. (i.e. it is not clear/compelling from the TLC image that BT4243 has activity)

-Please check the sugar key provided in Figure 4 – there is no blue circle in the key, although one is visible in structures in Fig 4A.

-There is an unprocessed citation to Miyazaki 2022 on line 269.

-Line 297, missing word: "...binding proteins AND enzymes are highly..."

-How quantitative is the SDS-PAGE/blotting method used for Fig 7B? Were there internal standards used to show consistent sample loading/staining?

-I don't think I see an in-text reference to Fig 7D?

-In the sections beginning on lines 317 and 347, there are many references to Fig 8, but no specific panels are mentioned – the figure contains nine distinct parts, so it would be much easier to find relevant data if they were referred to in-text. Please consider my other comments regarding organisation/ordering of the panels and references to them. Make sure every panel is relevant to the text, is cited in text in order, etc

-Again, a plea for consistency in protein naming: on lines 460-463, the peptidase is referred to as BT4244-M60L, BT4244, and then just 4244.

-Similarly please be consistent with using UK vs US spellings: e.g. see line 530 where the word characterised/characterized appears twice with different spellings.

-Spell out all abbreviations at first use. Check IPTG line 568, 'porcine gastric mucin' on line 712 (PGM used many times in the manuscript before that). Lines 685-686, 'Kan50' and 'Cm20' are not explained.

-Section beginning 625: specify why two different stains were used for samples.

-Consistency of terminology: SelenoMet, SeMet, selenomethionine, Semethionine, are all used at different points in the methods.

-Section beginning line 635: how were absorbance values converted into activity for the assays described?

-Section beginning line 729 describes your approach for finding PULs. Are these PULs findable in e.g. the PULDB database tool?

-In Supplementary Figure 1, it looks from these images like there was higher loading of BT4244 in the second gel than the first, since nothing is visible in the * lanes in the first gel. Can the authors comment on the consistency of concentration, loading, etc, since this figure is meant to show the specificity of the enzyme.

-Supplementary Figure 2C, the text is rather too small to be legible.

-Supplementary Figure 5 and corresponding Methods description – how did you distinguish cytoplasmic from periplasmic?

Reviewer #2

(Remarks to the Author)

The paper submitted by D.A. Ndeh et al. is an important study dealing with mucin degradation by a prominent human gut bacterium, *Bacteroides thetaiotaomicron*. This is a considerable piece of work, combining many in vitro approaches (quantitative binding assays, glycoside hydrolase assays using a large diversity of glycosides and enzymatic cascades, phosphorylation assays, kinetic studies), structural analyses, and, in the native strain, growth competition and complementation experiments with knock-out mutants, as well as immunofluorescence microscopy used for cellular localization of the studied proteins.

This major paper describes i) the deciphering of the function of a previously uncharacterized kinase (BT4240), showed to play a key role in GalNAc metabolism by Bt; ii) a new function (the α -GalNAc'ase activity displayed by BT4243) for members of the GH109 CAZy family; iii) a new crystallographic structure of a SusD-like protein (BT4246) involved in mucin catabolism, which shares less than 32% sequence identity with structurally characterized homologs; iv) more generally, the molecular dissection of the mechanisms involved in GalNAc containing host glycans.

I really enjoyed this manuscript, which represents a major contribution to a currently limited body of work on the molecular mechanisms involved in the microbiota-host interactions and gut homeostasis. Nevertheless, the minor points listed below should be addressed before publication of this paper.

- L184-185 : please explain the rationale of the comparison with CpGH89CBM32-5, PDB: 4AAX

- L208 : please indicate which protein is the closest structurally characterized homolog of BT4246-SusD-like. In the Discussion section, the interest of solving this new structure of a SusD-like protein should be highlighted.

- L290-294 : NMR experiments should allow the authors to be more conclusive on the phosphorylation position (O1 or O6), and thus to further discuss how phosphorylated GalNAc could be metabolized in the cytoplasm.

- L405-409 and in the Discussion: Although informative, the data presented Supplemental fig. 8 could be further discussed, especially regarding the health status of the individuals (in particular IBD vs healthy). Are the data consistent with the abundance of Bt in the microbiota of patients suffering from IBD? In addition, the criteria chosen for assessing the occurrence of the BT4240-50 PUL in human gut metagenome data seem not very stringent (E values $<1E-20$, and nucleotide identities $>90\%$ over a length >100 bp). Finally, the quantification of the prevalence and abundance of each gene of the BT4240-50 PUL (in particular of the key protein BT4240) in the human gut metagenome as a function of health status would probably be more informative.

- L516-523 and L524-529: the authors suggest that PUL BT4240-50 could play a role in the breakdown of non-colonic mucins, but they found this PUL highly abundant in the colonic metagenome. This appears inconsistent. Could the authors

analyse the prevalence of PUL BT4240-50 in other metagenomes than the colonic one ? What about the prevalence and abundance of Bt in other human microbiota ?

- L418-430 : please shorten this section, which is rather an introduction than a discussion.
- Figure 2 : Problem of legend (E & F)
- The text should be improved :
 - o missing dots e.g. L215
 - o missing comma e.g. L530
 - o missing spaces e.g. L84, L150, L217, L266, L313, L405, L758
 - o problems of references L269, L399, L619
 - o too long sentences that could be split to gain in clarity e.g. L54-59, L91-L94
 - o L71 inflammatory bowel disease should be inflammatory bowel diseases
 - o L91, L324: *in vitro* should be italic
 - o L116 and L207: for manuscript homogeneity, section titles should be summary points
 - o L153 : ...is also prominent in the gut...
 - o L510 : level expression of the operon BT4240-43
 - o L730 : for comparison from
 - o Throughout the text, galactose /fucose residues should be galactosyl /fucosyl residues
 - o L392 : Released GalNAc residues from various sources are imported

Reviewer #3

(Remarks to the Author)

The manuscript by Ndeh et al. revisits a previously identified and transcriptionally upregulated locus during the growth of *Bacteroides theta* on mucin O-glycans. Understanding the molecular mechanisms of mucin breakdown by model gut microbiota groups is of great interest and relevance to symbiosis and pathogenesis, both strongly influenced by mucin turnover.

A glycopeptidase from this locus has previously been studied in detail and was shown to cleave the backbone of mucins with a preference to the peptide bond adjacent to truncated T and Tn epitopes. The authors have performed biochemical, structural and microbial analyses of proteins encoded by this locus, which could be expressed in soluble form. The main aspects of novelty of the work are: 1) The fact that the glyco-peptidase is active on the IgA, which is abundantly secreted by the host to the lumen of the gut, 2) The ability of the GH109 GalNAcase to cleave the conjugated GalNAc of the Tn antigen, 3) The key role of the Kinase that phosphorylates the GalNAc in global GalNAc catabolism in *B. theta*.

The experiments are generally well designed and performed. Although the authors have highlighted novel aspects of this previously described locus, there are central questions regarding the proposed role of the locus, which remained unanswered. In my opinion, the impact of the work would be greatly increased if these issues would be addressed.

Major issues:

1. It is very interesting that the authors showed activity on IgA, abundantly secreted and found to coat several gut microbiota taxonomic groups. Although the biochemistry is solid, it remains unclear whether *B. theta* is able to cleave IgA under *in vivo* relevant conditions. At a minimum, the authors should demonstrate whether *B. theta* cells can perform the cleavage and if they are able on growth on IgA as a sole carbon/Nitrogen source. In addition, would it be possible for the authors to perform a (semi)quantitative measurement of the relative activity on IgA vs mucin? e.g. from western blot/immune staining,

2. I do agree with the authors' conclusions on mucin being the likely substrate for the locus, given the previously known activity of the glycopeptidase and the transcriptional data. Unfortunately, the binding analyses of the transport protein are performed on monosaccharides and not on Tn antigen or T antigen glycopeptides, which are the substrates of the glycopeptidase and the most likely uptake ligands for the SusD transport protein. Similarly, the crystallography is performed on released O-glycans and not on O-glycopeptide products of the protease.

The model in Fig. 9, albeit plausible, is not supported by the experimental data. The uptake of glycopeptides is not demonstrated, and if glycopeptides are taken up, it is unclear what is the size interval or preference, which could be internalized by the transport protein. This is key data since the authors, correctly so, point out that T and Tn antigens are less common in colonic mucin. Somehow, the peptide fate is not addressed in the model Fig. 9 and the peptide seems to disappear after glycosylation from the figure and its legend text. It would be splendid if the authors could address this key point, by demonstrating the uptake of the products of the glycopeptidase. If this cannot be achieved, then the text, figure, and figure legends need to be toned down and claim modified appropriately.

3. It is great that the authors obtained a crystal of SusD in complex with the O-glycans from a complex pool. This protein is expected to internalize the monosaccharide conjugated to a glycopeptide. I am perplexed that the C1 of the GalNAc moiety is pointing out to the solvent away from the binding site in contrast to the binding of glycopeptides in glycopeptidases, where the peptide chain is bound to the binding site and the sugar ring is protruding towards the solvent. How does the transport protein discriminate between a free GalNAc and a glycopeptide if the peptide chain is distant from the surface of the protein? Is it possible to do some modelling or seem whether the GalNAc can be flipped so the C1 is pointing towards the bottom of the pocket/groove? Perhaps this merits to be at least discussed in the presentation of the crystallography results.

Minor:

Title: mucin-O-glycans: please italicize the O here and elsewhere in the manuscript per formal chemical nomenclature
N-acetylgalactosamine: please italicize the N here and elsewhere.

L32, "mucin O-glycan glycoprotein", repetitive omit glycan and write "O-glycoprotein"

L49, "rigid and impenetrable": compact may be more precise. Not sure if rigidity per se has been reported, and impenetrable is only valid for bacteria not smaller metabolites! Please rephrase for clarity.

L69: Not sure what is meant by "must be coordinated". Perhaps balanced would be a better wording, as fiber starvation triggers increased mucin breakdown and loss of balance, so there is no coordination as such, just a balance, that can be disrupted by dietary or clinical factors. In fact these three words could be omitted altogether, since the balance is clarified in the next sentence

L71, consider using systemically instead of "distally"

L102, "seen exhibit": Lingo, seems to lack a "to" or something

Results:

L131, why is Sus in italics?

L140, check lingo please, "surface that" appears wrong, swap to which

L141: Please define "small oligomers", i.e. indicate a DP, it has been shown that such proteins target large oligomers more efficiently than smaller counterparts.

L150: Can the authors comment on why there is still some signal in the KO staining? The polyclonal antibodies are expected to target the CBM32, which is joint to many proteins from this bacterium including the target SGBP? Have the authors investigated the level of cross-reactivity of the antibody to robustly argue against the signal being at least in part stemming from other CBM32 containing proteins, which may be induced by the complex substrate? Please comment, and maybe indicate shared aa % identity.

L157, the Fab α and the Fc α should be labelled in the figure and their sizes indicated to make it clear that the cleavage fits the size of the domains in the cited Fig. 2F panel.

L177-178, the K_a is lower on the GalNAc, so what is the basis to claim preference for the amino sugar?

L183-183, please mention which structures are AlphaFold models to distinguish them from experimental crystallography models so that is clear to the reader.

L192. Galactose-configured may be clarified to "..with a terminal Gal"

L211-213, no Fig. panel is cited for the density, so please cite the right panel.

L217, insert space between protein(Fig. 3F)

L235, the k_{cat} appears low, would be interesting to try a galactobiose b-Gal1,3Gal, to assess of the enzymes efficiency is better than on the HexNAc?

L247, L257-259, Interesting with the activity on the Tn antigen. However, the release of the GalNAc is not obvious from the TLC in Fig.4D, which lacks a GalNAc standard, please clarify or even better include the GalNAc as a standard to make this clear to the reader. Is this activity comparable to the activity on the blood group A antigen? Or is this a minor trace activity? Would be splendid if enzyme concentration, incubation temperature, and time are mentioned in the legends of this and all similar figures to evaluate the levels of turnover without referring to the methods.

L262, Space between genus and species

L269, fix ref!

Figs.

Fig. 1A: Please make sur that there is key in the figure or legend to all things indicated. The different cell types, sulphate linkages, mucin domain are all not defined, please clarify this figure for the reader.

Fig. 2A: Is the cartoon representation of the enzymes under the gene locus drawn in scale?. Please clarify in the legend.

Fig.2B,C please label the markers with the M_w of the different bands for the Coomassie and the western staining, so the reader can appreciate the size of the band, not any obvious where to look in the Coomassie stained gel. The resolution is not good or the same in the C panel, so imperative with the labelling of the marker bands!

Fig. E. Neat experiment, would have been nice to see the marker and to indicated the expected size of the fragment though.

Fig. 3: The use of colors is a bit inconsistent in this figure. Why are the carbon atoms in black in panels D and E and blue in F and G. Please color only the carbon atoms and not all atoms in panel C, i.e. maintain the elemental color code to make the chemistry of the side chain and ligand easier to spot for the reader.

Fig. 7A: The black arrow which assigns of the cleavage site of the glycopeptidase is confusing. This protease prefers de-sialylated Tn/T antigens?. Similarly, why has the arrow of the sialidase been placed on the 2-3 sialyl? I guess this sialidase is also active on other linkages? Please correct/clarify.

Version 1:

Reviewer comments:

Reviewer #1

(Remarks to the Author)

The authors of this manuscript have done an admirable job of addressing the comments from all reviewers. I appreciate them spending time to focus on aspects I highlighted that improve the readability of the work.

In your response to my comments I think you mixed up operon 1 and operon 2, but this seems to be correct in the actual paper.

Thank you also for clarifying the TLC aspect, although I think that there is a way to optimise for staining of amino sugars. The aniline stain you use has been shown to work well to stain GlcNAc for example, but different running buffers (replacing the acetic acid with ethanol) can also be used that give better resolution with amino sugars and charged sugars.

Reviewer #3

(Remarks to the Author)

I would like to thank the authors for the great job revising the manuscript, which is reads much better.

I have two issues which might be important to address in the final revision:

1) L432-434: The sentence seems to imply that SusD recognizes the terminal Gal, but is indiscriminate to the rest of the glycopeptide. As stated in my first revision, the claim is not warranted based on the crystal soaks with oligosaccharides, which are not relevant substrates. The model added in the revision (SI Fig. 4) unfortunately does not add support to the mode of binding of glycopeptides as the glycopeptide is not modelled/docked. Therefore, I strongly advise the authors to nuance this claim, which is overshadowed by the lack of explanation for the mode of selection of a preferred size range of glycopeptides for uptake by the transport system. Maybe substitute with a sentence à la "The recognition of the terminal Gal is key..., but additional interactions with the SusC or SusD may facilitate the selection of preferred uptake glycopeptides." Or something along these lines.

2) L479-481: "While these findings... may play a role in UC"

No causative links are shown, but merely associations. Please tone down to what is permissive by the data, which finds no consistent or compelling support based on the following discussion.

Please have a look at a few minor issues in the revised version:

L32: Please correct to mucin-O glycoproteins to "mucin O-glycoproteins"

The abstract could benefit for a round of focusing and shortening.

L241: Add Å to the RMSD.

Reviewer #4

(Remarks to the Author)

The authors have done an excellent job incorporating the suggested experiments and addressing all concerns. The additional data and clarifications strengthen the study, making the findings even more robust. The explanation for not including NMR is also well-reasoned and clearly justified.

With these revisions, the manuscript is now clear, well-structured, and suitable for publication in Nature Communications.

REVIEWER COMMENTS

Reviewer #1 (Remarks to the Author):

Ndieh et al have used a combined genomic, structural, and biochemical approach to characterise the mechanism of action of a key mucin-targeting PUL from *Bacteroides theta*, regarded as a model species for studying the human gut microbiome and a cornerstone of that community. The experimental work has been performed to a high quality and I agree with the conclusions the authors draw from their work. Although the enzyme characterisation results are not surprising, and align well with the predicted functions of the respective proteins, it is interesting to see the data put together and contextualised. Thus, the majority of my comments relate to the presentation of the data, as there are unfortunately a few issues here. There are inconsistencies and lacks of clarity in the text and figures, which undermine the quality of the work. See some specific issues described below, most of which should be straight-forward to address.

We thank the reviewer for their supportive and constructive comments about the manuscript. We have tried to make all of the revisions requested – hopefully the MS is much clearer now.

-Consider using the more up to date term Bacteroidota instead of Bacteroidetes?

- Ok changed throughout MS.

-Line 59, “Central to all mucin O-glycosylated sugars...” Should this be “Central to all mucin O-glycosylated proteins...”?

- Changed as suggested

-Figure 1, it's a bit unclear which elements belong to panel A.

- The figure has been modified to make this clearer.

-You mention in the Introduction and elsewhere that the PUL is likely involved in metabolising GalNac-glycans other than mucin, and the enzymatic evidence for this is clear, but it is not clear to me if it is known whether the whole PUL is highly expressed during growth on other substrates? I apologise if this is mentioned in the text, but if so, it is not clearly presented.

While the whole PUL is not expressed on non-mucin substrates, we have mentioned at the start of the results section (Line 118) that operon 2 containing the GalNac kinase BT4240 and other GHs is expressed at a relatively higher basal level, suggesting the kinase is present in the cell at all times and primed to respond to GalNac from a range of glycan sources.

-On a similar note, the Introduction states that one of the two operons in the PUL shows higher basal expression than the other. As far as I can tell, the one with higher basal expression is the one encoding the CAZymes – the Discussion does not seem to come back to this point of there being two separately controlled operons but is interesting and perhaps relevant to my previous comment.

Indeed this is related to the previous comment as it appears that the kinase and GH part of the locus (operon 2) are constitutively expressed (i.e. a relatively higher basal level, although they are still upregulated by mucins). In contrast the operon 1 (the surface SusCD apparatus and the glycoprotease) is more tightly controlled and only really responds to mucins.

The constitutive nature of operon 2 (kinase and GHs) suggests that the enzymes are involved in diverse processes. Thus, while the operon 1 is highly specific for mucins, the constitutive part (i.e. the kinase and GHs) are involved in the metabolism of other non-mucin glycans – with the example of the kinase also being key for CS metabolism as demonstrated here. In addition, the GHs could be involved in the processing of other alpha-GalNAc containing glycans (both mucin and non-mucin). This possibility is discussed in the original MS on lines 566-575

-Line 118, word missing “...encoded proteins are involved IN the acquisition and breakdown...”

Corrected.

-The phrases *in silico*, *in vivo*, and *in vitro* should always be in italics but should never be connected by hyphen (e.g. *in-silico*), please correct throughout.

Corrected.

-There are issues with the ordering/organisation of items in several figures (Fig 1, Fig 2, Fig 3, Fig 8...). Please assume that the reader will look for panels A, B, C, etc in logical reading order, i.e. left-to-right and then top-to-bottom. Figure 3 especially is hard to comprehend, since the panels are not presented in the alphabetical order to which they are referred in the caption, so it feels chaotic to try to read the figure.

Apologies for the confusion. The figures have now been changed as suggested to make these clearer.

-In Fig 2, I think panel F is mislabelled as E? And no panel is labelled F, despite the caption and an in-text reference to Fig 2F.

This has now been corrected and labelled appropriately. Please see new Fig. 2.

-Line 163 – where does this lectin come from?

GalNAc-binding lectin from *Helix aspersa* (Sigma cat. L8764). This has been added to Methods (line 818).

-Line 171 mentions a BACON domain of unknown function – since the BACON name derives from “Bacteroides-associated carbohydrate-binding often N-terminal” I think it is wise to explain the nomenclature a bit more. Why is it called a BACON if it is not expected to bind carbohydrates?

A carbohydrate binding function was originally predicted based on bioinformatic analyses but to date this has not been experimentally demonstrated (Mello et al., 2010: PMID: 20416301). Recent structural and bioinformatic data suggests it may be a linker in surface anchored proteins, especially those that are components of Bacteroidota outer membrane utilosomes (White et al. 2024 PMID: 37286596; Hameleers et al., 2024, PMID: 38735878 Larsbrink et al., 2014, PMID: 24463512). This text has been modified to clarify this point (lines 177-180). We also have experimental data demonstrating that the BACON domain is not a CBM (see next point).

-Line 178 and 179 refer the reader to Suppl Fig S2 to see binding data for the CBM32 and BACON constructs, but the data for the BACON alone are not shown. Also, is “BACON-CBM32” the same as “BT4245-FL”? Please use consistent naming conventions across text, figures, and captions – this is a recurring issue throughout the paper and makes it difficult to keep track of or understand the results.

The BACON-CBM32 refers to the domains from BT4244 not BT4245 SGBP. This has now been clarified in the main text and the notes to the ITC data Table (S1). The BACON alone data is now shown in Table S1 and an example ITC in Suppl Fig 2.

-Line 185, should this not refer to ‘ligand’ recognition rather than ‘substrate’, since it is a CBM being discussed?

Corrected.

-Line 200, word missing, “...partner SusC-like PROTEINS for active...”

Corrected.

-As mentioned above, the layout of Figure 3 is chaotic and hard to follow. Moreover, I can find no reference at all to panels A-C of the figure in the actual manuscript. Why include those images if they are not relevant to the paper? Please make sure to refer to all figures.

Figure has been edited to improve clarity, but it was originally referenced in the text (lines 187-188 – see below).

184	alignment of the CBM32 domains of BT4244, BT4245 and the C. perfringens GH89 enzyme
185	(CpGH89CBM32-5, PDB: 4AAX) showed spatial conservation of key residues implicated in substrate
186	(GalNAc) recognition in CpGH89CBM32-5 including H1392 and N1428 and F1483 corresponding to
187	residues N352, H316 and F417 in BT4245-SGBP and H168, N202 and F260 in BT4244-CBM32 (Fig
188	3A, B, C). H1392 and N1428 make polar hydrogen bond interactions with the GalNAc ligand [53] in
189	agreement with data from site directed mutagenesis and subsequent ITC analyses which showed that

-Figures 3D and 3E are perhaps too small to see the ligand clearly.

Fig 3 has now been rearranged and edited for clarity and the BT4246 structure as well as the ligand binding site and ligand is hopefully now clearer. The old panel E (comparison of BT4246 SusD and the canonical starch binding SusD) has now been moved to Suppl Fig 3B to provide space to allow comparison of the ligand binding sites in more detail.

-On a similar note to an earlier comment, the sections of figure 4 are not cited in order (Fig 4A line236, Fig 4B line242, Fig 4F line244, Fig 4C line247, Fig 4D line253). Ideally you could rethink the order of panels in the figure so they can be cited in order. Unless I missed it, there is no reference to Fig 4E anywhere in the text.

Sections of Fig. 4 have now been cited in order and figures also re-arranged to ease reading. Data in original Fig 4E (now Fig 4F) has also now been referenced in the text and cited in order (line 270-278).

-Figure 4C is not very compelling if intended to show a different product profile for one vs two enzymes combined. (i.e. it is not clear/compelling from the TLC image that BT4243 has activity. We apologise for the lack of clarity in Fig 4C (now 4D) but have now modified the figure and added info to the legend to make clear what is happening. The Gal released by the GH2 (BT4241) co-migrates on the TLC with GalNAc released by BT4243 and as the N-acetyl sugar doesn't stain as well as Gal this masks the GalNAc. The original image that contained the GalNAc standard was cropped, which excluded the GalNAc and Tn antigen standards. It has now been uncropped in 4D to show the standards and the evidence of co-migration (or similar retention times) for Gal and GalNAc. Direct evidence of GalNAc'ase activity of BT4243 in Fig 4D is therefore seen with the disappearance of the GalNAc-Ser substrate (Tn antigen, Fig 4D) rather than appearance of the GalNAc product. This is now explained in the legend. Further evidence of BT4243 activity is also demonstrated in Fig 4E against BSM, which shows clear release of GalNAc by the enzyme after the mucin has been de-silylated to uncap the Tn structures. In addition, BT4243 activity vs BSM and Tn is also seen in Fig 5 and vs IgA in Fig 7B as shown by the lack of anti-GalNAc lectin staining in the BT4243 treated IgA1.

-Please check the sugar key provided in Figure 4 – there is no blue circle in the key, although one is visible in structures in Fig 4A.

Corrected.

-There is an unprocessed citation to Miyazaki 2022 on line 269.

This has now been processed (PMID: 34826537) and added to ref # 63 (line 1206)

-Line 297, missing word: "...binding proteins AND enzymes are highly.."

Corrected.

-How quantitative is the SDS-PAGE/blotting method used for Fig 7B? Were there internal standards used to show consistent sample loading/staining?

Quantification was done by densitometry analyses which although not 100% accurate was able to provide data for comparative analyses. Data was collected from repeat experiments and were in agreement with data from previous studies by Noah et al., 2017 showing a decrease in

BT4244 protease activity with sialylation and increased protease activity upon de-sialylation, and de-galactosylation

-I don't think I see an in-text reference to Fig 7D?

Corrected – now included.

-In the sections beginning on lines 317 and 347, there are many references to Fig 8, but no specific panels are mentioned – the figure contains nine distinct parts, so it would be much easier to find relevant data if they were referred to in-text. Please consider my other comments regarding organisation/ordering of the panels and references to them. Make sure every panel is relevant to the text, is cited in text in order, etc

Apologies for this confusion this has now been corrected in the revised MS (see lines 358-87).

-Again, a plea for consistency in protein naming: on lines 460-463, the peptidase is referred to as BT4244-M60L, BT4244, and then just 4244.

Now corrected – changed everything in the text to BT4244-M60L and this is first defined in the Introduction.

-Similarly please be consistent with using UK vs US spellings: e.g. see line 530 where the word characterised/characterized appears twice with different spellings.

Corrected - all words were changed to UK spellings.

-Spell out all abbreviations at first use. Check IPTG line 568, 'porcine gastric mucin' on line 712 (PGM used many times in the manuscript before that). Lines 685-686, 'Kan50' and 'Cm20' are not explained.

- IPTG line written out in full first time used.
- PGM defined as 'porcine gastric mucin' first time used and then abbreviated to just PGM after.
- Kan50 and Cm20 explained.

-Section beginning 625: specify why two different stains were used for samples.

This was stated in line 628-29 in original article (now line 764-65) and it is because DPA helps with the detection of sialic acids and other N-acetylated sugars like GalNAc which are generally not very visible with orcinol. Please see snipped text below from original MS. This has been made clearer in the Methods of the revised MS.

624 **Glycoside hydrolase and sugar kinase assays.**

625 *Thin layer chromatography (TLC)*. TLC analysis of sugars was performed using silica TLC plates
626 (Sigma) as previously described [65]. Reactions (3-6 µl of each) were spotted on to plates and samples
627 resolved in butanol/acetic acid/water buffer (2:1:1) in glass tanks. Plates were then air-dried and dipped
628 in either orcinol/sulfuric acid (0.5% (w/v) orcinol in 10% sulfuric acid) or diphenylamine–aniline–
629 phosphoric acid (DPA, used to detect sialic acid [83]) developers before being dried at 70°C for up to
630 10 min to visualize sugars. Reactions (50 µl total) were set up in 1.5 ml microcentrifuge tubes containing
631 enzyme (BT4241 or BT4243 at 0.3-1.2 µM) and substrate (5-10 mM disaccharides or 4 mg/ml mucins)

-Consistency of terminology: SelenoMet, SeMet, selenomethionine, Semethionine, are all used at different points in the methods.

Corrected all to SelenoMet after initial abbreviation.

-Section beginning line 635: how were absorbance values converted into activity for the assays described?

Absorbances were converted to enzyme activity (mols product released) using the molar extinction coefficient of NADH at 340 nm (6220 M⁻¹ cm⁻¹). Initially we used an extinction coefficient of 4500 M⁻¹ cm⁻¹ for PNP based on other studies but have performed our own in-house measurements in Tris/NaCl buffer pH 8.0 and MOPS buffer pH 7.0 which gave extinction coefficient values of 16,800 M⁻¹cm⁻¹ and = 9,000 M⁻¹ cm⁻¹ respectively. These new values were used to recalculate the enzyme kinetics and the new results have been used to update table S3. We thank the reviewer for the drawing our attention to this.

Section beginning line 729 describes your approach for finding PULs. Are these PULs findable in e.g. the PULDB database tool? Confirm that strains listed are in PULDB and cite PULDB

We identified these PULs in both PULDB and the IMG/M database. The figures were generated with the gene neighbourhood tool in the IMG/M database. The entries in Supplementary figure 12 (metagenome analysis) correspond to the following entries in the PULDB:

Bacteroides thetaiotaomicron VPI-5482: BT4240-50 --- Literature-derived PUL 78

Bacteroides faecis MAJ27: KCYDRAFT_01771-81 --- Predicted PUL 70 (Note *B. faecis* strain 23150 is in PULDB, proteins range %ID values compared to MAJ27: 99.82%-100%)

Bacteroides caccae ATCC43185: BACCAC_1835-46 --- Predicted PUL 34 in PULDB

Reference

IMG/M: Chen, I.M.A., Chu, K., Palaniappan, K., Ratner, A., Huang, J., Huntemann, M., Hajek, P., Ritter, S., Varghese, N., Seshadri, R. and Roux, S., 2021. The IMG/M data management and analysis system v. 6.0: new tools and advanced capabilities. *Nucleic acids research*, 49(D1), pp.D751-D763.

PULDB: Terrapon, N., Lombard, V., Drula, E., Lapébie, P., Al-Masaudi, S., Gilbert, H.J. and Henrissat, B., 2018. PULDB: the expanded database of polysaccharide utilization loci. *Nucleic acids research*, 46(D1), pp.D677-D683.

This information has now been added to the revised MS.

-In Supplementary Figure 1, it looks from these images like there was higher loading of BT4244 in the second gel than the first, since nothing is visible in the * lanes in the first gel. Can the authors comment on the consistency of concentration, loading, etc, since this figure is meant to show the specificity of the enzyme.

We agree that the BT4244 bands on the second gel look somewhat more intense than on the first gel, however, the same amount of enzyme was added to all samples and the same volume also loaded on the gels. We assume the slightly higher apparent loading of BT4244 on the second gel is due to variability in staining. This explanation is supported by the markers which are slightly darker on the second gel even though the same volume was used on both. Thus the main conclusion from these gels that the enzyme is only active on IgA1 is not affected.

-Supplementary Figure 2C, the text is rather too small to be legible.

Text has been removed from the figure and explained in legend and the figure has been split and made larger in revised MS – see new Supplemental fig. 2, 3.

-Supplementary Figure 5 and corresponding Methods description – how did you distinguish cytoplasmic from periplasmic?

For the kinase i. the lack of signal peptide ii. kinase needs ATP for activity and ATP is not in the periplasm iii. BT4240 is absent in membrane fractions (and detected mainly in soluble fraction) showing no interaction or protein in transit. For BT4243 and BT4241, both have Type I signal peptides and are thus most likely secreted into the periplasm in *Bacteroides spp*, but also our results show evidence for these enzymes in membrane fractions, supporting their transport across the cytoplasmic membrane into the periplasm. Also notice that all the surface exposed proteins with SPII signal are often detected in the medium but this was not the case for BT4243 and BT4241.

How these predictions were made is described in the ‘Cellular localisation’ part of the Results section and has been added to the legend to Supp Fig 9 in revised MS for clarity.

Reviewer #2 (Remarks to the Author):

The paper submitted by D.A. Ndeh et al. is an important study dealing with mucin degradation by a prominent human gut bacterium, *Bacteroides thetaiotaomicron*. This is a considerable piece of work, combining many in vitro approaches (quantitative binding assays, glycoside hydrolase assays using a large diversity of glycosides and enzymatic cascades, phosphorylation assays, kinetic studies), structural analyses, and, in the native strain, growth competition and complementation experiments with knock-out mutants, as well as immunofluorescence microscopy used for cellular localization of the studied proteins.

This major paper describes i) the deciphering of the function of a previously uncharacterized kinase (BT4240), showed to play a key role in GalNAc metabolism by Bt; ii) a new function (the α -GalNAc’ase activity displayed by BT4243) for members of the GH109 CAZy family; iii) a new crystallographic structure of a SusD-like protein (BT4246) involved in mucin catabolism, which shares less than 32% sequence identity with structurally characterized homologs; iv) more

generally, the molecular dissection of the mechanisms involved in GalNAc containing host glycans.

I really enjoyed this manuscript, which represents a major contribution to a currently limited body of work on the molecular mechanisms involved in the microbiota-host interactions and gut homeostasis. Nevertheless, the minor points listed below should be addressed before publication of this paper.

We thank the reviewer for their kind comment about the manuscript.

- L184-185 : please explain the rationale of the comparison with CpGH89CBM32-5, PDB: 4AAX

CpGH89CBM32-5 is a well-characterised Gal/GalNAc binding CBM32 domain of CpGH89 and is one of the few CBM32s in the PDB with GalNAc bound and hence useful for comparison to identify GalNAc interacting residues in the *B.theta* CBM32s. This has now been stated in the text in lines 192-193

- L208 : please indicate which protein is the closest structurally characterized homolog of BT4246-SusD-like. In the Discussion section, the interest of solving this new structure of a SusD-like protein should be highlighted.

Via a DALI search, BT3013 is the closest determined structure to BT4246, and we have included a brief comparison of these proteins in the SusD structure section of the results (and new Supplemental Fig 5). BT3013 is encoded within a PUL that is upregulated during in vitro growth on porcine mucosal glycans, but nothing more is known about the ligand target of this protein. However, for BT4246 we provide an explanation as to how capture of the non-reducing end of galactose may perhaps allow the promiscuous capture and transport of peptide-linked mucosal glycans (see also new Supplemental Fig 4). We have also provided some info in the Discussion regarding the interest in the SusD structure as requested (revised MS lines 243-45).

- L290-294 : NMR experiments should allow the authors to be more conclusive on the phosphorylation position (O1 or O6), and thus to further discuss how phosphorylated GalNAc could be metabolized in the cytoplasm.

We agree that this is an interesting point and performed some experiments and structural analyses to determine which of the O1 or O6 sites in the N-acetyl sugars is the target site for phosphorylation by BT4240 kinase. To confirm this, we used high performance anion exchange chromatography (HPAEC) and commercially available GlcNAc phosphate standards GlcNAc-1-phosphate (G1P) and GlcNAc-6-phosphate (G16P) due to the limited availability of relevant phosphorylated GalNAc sugars to use as standards. Both sugars had different retention times by HPAEC analyses and the product generated following treatment of GlcNAc with BT4240 was confirmed to be G1P and not G6P (Fig. 6E). Hence BT4240-kinase phosphorylates at O1 of the N-acetyl sugar. The text has been updated with this new information (lines 322-324). Additionally, the closest functionally characterised homologue of BT4240, the HexNAc kinase from *Bifidobacterium longum* BL1642 (39% identity) phosphorylates at O1 and comparison of the AlphaFold model of BT4240 with BL1642 reveals that the *B. theta* enzyme contains very similar sugar (and ATP) binding sites supporting O1 being the site of GalNAc phosphorylation in BT4240. As both substrate binding sites are so conserved we felt it wasn't necessary to carry out the NMR experiments to experimentally confirm this finding, especially with the additional costs this would ensue. We have added this new information to the paper and a figure to show the similarities between the kinases – see new Supplemental Fig. 5 and the kinase activity description in the revised MS results (lines 322-324).

- L405-409 and in the Discussion: Although informative, the data presented Supplemental fig. 8 could be further discussed, especially regarding the health status of the individuals (in particular IBD vs healthy). Are the data consistent with the abundance of Bt in the microbiota of patients suffering from IBD? In addition, the criteria chosen for assessing the occurrence of the BT4240-50 PUL in human gut metagenome data seem not very stringent (E values <1E-20, and nucleotide identities >90% over a length > 100 bp). Finally, the quantification of the prevalence and abundance of each gene of the BT4240-50 PUL (in particular of the key protein BT4240) in the human gut metagenome as a function of health status would probably be more informative.

We thank the reviewer for having raised this interesting point. Consistent with the literature that indicates that Bt can suppresses inflammation in the context of IBD, a Chi square test across the analysed HMP and MetaHit metagenomic datasets we considered (Supplementary Figure 11), indicated that the 3 PULs, BT4240-50 and the two BT4240-50-like PULs from Bc and Bf, are underrepresented among (i) IBD patients (combining UC and CD patients, p-value 0.0065) or (ii) among UC individual subsets of patients (p-value 0.0349) compared to healthy controls. However, (iii) there was no significant underrepresented of these PUL among CD patients (0.0947). This information has now been added to the revised MS, lines 470-503

- L516-523 and L524-529: the authors suggest that PUL BT4240-50 could play a role in the breakdown of non-colonic mucins, but they found this PUL highly abundant in the colonic metagenome. This appears inconsistent. Could the authors analyse the prevalence of PUL BT4240-50 in other metagenomes than the colonic one ? What about the prevalence and abundance of Bt in other human microbiota ?

It is true that Bt is normally considered a member of the colonic microbiota as it is present in stool sample, but the potential role of the PUL in breakdown of non-colonic mucins was based on the fact that mucins produced further up the digestive tract (e.g. oral, respiratory, stomach etc) will end up in the colon as they transit the GIT. Thus bacteria in colon will likely have adapted to use both colonic, but also upper GI tract/respiratory mucins.

This point was made on lines 518-520 ‘...thus PUL BT4240-50 could play an important role in the breakdown of these non-colonic mucins as they converge in the colon, somewhat mirroring exposure of B. theta to dietary glycans.’ (now line 613-614)

- L418-430 : please shorten this section, which is rather an introduction than a discussion.

This has now been shortened as suggested.

- Figure 2 : Problem of legend (E & F)

This has now been corrected.

- The text should be improved :

o missing dots e.g. L215 – corrected.

o missing comma e.g. L530 –corrected.

o missing spaces e.g. L84, L150, L217, L266, L313, L405, L758 corrected.

o problems of references L269, L399, L619 doesn't match (corrected)

o too long sentences that could be split to gain in clarity e.g. L54-59, L91-L94 - corrected.

o L71 inflammatory bowel disease should be inflammatory bowel diseases – corrected.

o L91, L324: in vitro should be italic – corrected.

- o L116 and L207: for manuscript homogeneity, section titles should be summary points – All headings now summary points.
 - o L153 : ...is also prominent in the gut...- corrected.
 - o L510 : level expression of the operon BT4240-43 - corrected.
 - o L730 : for comparison from - corrected.
 - o Throughout the text, galactose /fucose residues should be galactosyl /fucosyl residues - corrected
 - o L392 : Released GalNAc residues from various sources are imported – corrected.
-

Reviewer #3 (Remarks to the Author):

The manuscript by Ndeh et al. revisits a previously identified and transcriptionally upregulated locus during the growth of *Bacteroides theta* on mucin O-glycans. Understanding the molecular mechanisms of mucin breakdown by model gut microbiota groups is of great interest and relevance to symbiosis and pathogenesis, both strongly influenced by mucin turnover.

A glycopeptidase from this locus has previously been studied in detail and was shown to cleave the backbone of mucins with a preference to the peptide bond adjacent to truncated T and Tn epitopes. The authors have performed biochemical, structural and microbial analyses of proteins encoded by this locus, which could be expressed in soluble form. The main aspects of novelty of the work are: 1) The fact that the glyco-peptidase is active on the IgA, which is abundantly secreted by the host to the lumen of the gut, 2) The ability of the GH109 GalNAcase to cleave the conjugated GalNAc of the Tn antigen, 3) The key role of the Kinase that phosphorylates the GalNAc in global GalNAc catabolism in *B. theta*.

The experiments are generally well designed and performed. Although the authors have highlighted novel aspects of this previously described locus, there are central questions regarding the proposed role of the locus, which remained unanswered. In my opinion, the impact of the work would be greatly increased if these issues would be addressed.

Major issues:

1. It is very interesting that the authors showed activity on IgA, abundantly secreted and found to coat several gut microbiota taxonomic groups. Although the biochemistry is solid, it remains unclear whether *B. theta* is able to cleave IgA under in vivo relevant conditions. At a minimum, the authors should demonstrate whether *B. theta* cells can perform the cleavage and if they are able on growth on IgA as a sole carbon/Nitrogen source. In addition, would it be possible for the authors to perform a (semi)quantitative measurement of the relative activity on IgA vs mucin? e.g. from western blot/immune staining,

This is an interesting point and to address the reviewers question we cultured *B. theta* cells in PGM or glucose together with a small amount of IgA1 (unfortunately IgA is prohibitively expensive for us to use as the sole C-source) overnight and the next day collected supernatants and analysed by SDS PAGE. Our results showed some processing of IgA had occurred during growth on PGM (and to a lesser extent on glucose), suggesting that *B. theta* cells are able to degrade the immunoglobulin in culture (see Supplemental Fig. 1C). This info has been added to the revised MS (see line 170-172). Given the highly complex nature of the gut community, the impact of polymicrobial activity on IgA metabolism in *B. theta* also needs to be taken into consideration as other microbes may modify IgA facilitating its degradation by other microbes. The highly complex nature of mucins and its smeary nature on blots (Ref# 32) also makes quantification very difficult.

2. I do agree with the authors' conclusions on mucin being the likely substrate for the locus, given the previously known activity of the glycopeptidase and the transcriptional data. Unfortunately, the binding analyses of the transport protein are performed on monosaccharides and not on Tn antigen or T antigen glycopeptides, which are the substrates of the glycopeptidase and the most likely uptake ligands for the SusD transport protein. Similarly, the crystallography is performed on released O-glycans and not on O-glycopetide products of the protease.

We agree with the reviewer that it would be ideal to test the SusD against glycopeptides, these are not commercially available. However we have tested the SusD vs a range of disaccharides including core 1 (Fig S2 and Table S1) and shown the protein binds this preferentially over the component monosaccharides, LNB (GlcNAc in place of GalNAc), or b1,4 linked disaccharides, supporting preferential targeting of a terminal b1,3 linked Gal. Furthermore, in the crystal structure obtained from the O-glycan soaks there is additional density observed from the O1 of the Gal, but this does not interact with the protein suggesting that even if these were glycopeptides, the SusD doesn't actually recognise the peptide backbone (Fig 3E).

In addition, the localisation of the PUL encoded GHs in the periplasm supports deglycosylation of BT4244 glycopeptide products after transport across the OM, requiring SusD to bind to glycopeptides. We have also added a figure showing the proposed model for BT4246-4247 SusCD transport of Gal terminated glycopeptides (new Supplemental Fig 4).

The model in Fig. 9, albeit plausible, is not supported by the experimental data. The uptake of glycopeptides is not demonstrated, and if glycopeptides are taken up, it is unclear what is the size interval or preference, which could be internalized by the transport protein. This is key data since the authors, correctly so, point out that T and Tn antigens are less common in colonic mucin. Somehow, the peptide fate is not addressed in the model Fig. 9 and the peptide seems to disappear after glycosylation from the figure and its legend text. It would be splendid if the authors could address this key point, by demonstrating the uptake of the products of the glycopeptidase. If this cannot be achieved, then the text, figure, and figure legends need to be toned down and claim modified appropriately.

We have provided experimental evidence (enzymatic / cellular localisation/structural) for several parts of the model. However, we agree that we haven't shown direct evidence of glycopeptide uptake but we based this on the following lines of evidence:

1. The surface exposed glycopeptidase BT4244-M60L has been shown by us and others to cleave mucins and mucin like O-glycopeptides in an endo like fashion preferentially at the T and Tn structures which would release short Gal/GalNAc terminated glycopeptides at the cell surface (we show the enzyme is surface located). The protease also acts on glycoproteins including IgA generating glycopeptides
2. The SusD binds to terminal Gals in mucin-derived oligosaccharides (from crystal structure) as well as disaccharides (by ITC) and from the structure looks as if this binding mode could be a good way to target glycopeptides with variable peptide sequence as this peptide part would not need to interact with the SusD (see point 1 above and 3 below).
3. The PUL encoded GHs are periplasmic and act on the Gal/GalNAc terminated glycopeptides and hence these glycopeptide substrates must be imported into the cell to be processed. For example, the BT4243 enzyme which acts on the O-glycosidic bond between serine amino acid and GalNAc (in GalNAc α 1-O-Ser) and also in bovine submaxillary mucins is

located to the intracellular environment of the cell implying that this substrate must be imported into the cell to be processed

We think these data overall strongly support the model we propose, but agree that there may be other possibilities as to the target structures and so have toned down the text and figure/figure legend to address this. The fate of the released peptide is not known and we have indicated this with a question mark on the peptide in the figure

3. It is great that the authors obtained a crystal of SusD in complex with the O-glycans from a complex pool. This protein is expected to internalize the monosaccharide conjugated to a glycopeptide. I am perplexed that the C1 of the GalNAc moiety is pointing out to the solvent away from the binding site in contrast to the binding of glycopeptides in glycopeptidases, where the peptide chain is bound to the binding site and the sugar ring is protruding towards the solvent. How does the transport protein discriminate between a free GalNAc and a glycopeptide if the peptide chain is distant from the surface of the protein? Is it possible to do some modelling or seem whether the GalNAc can be flipped so the C1 is pointing towards the bottom of the pocket/groove? Perhaps this merits to be at least discussed in the presentation of the crystallography results.

The reviewer is correct in that the orientation of galactose bound to BT4246 is almost reverse that of what is seen in glycopeptidases. However, the orientation of the galactose is unambiguous in the electron density and the sugar cannot be modelled in a different orientation; this becomes most apparent at the C6-O6 position. However, we think this orientation of galactose makes sense in that it allows for capture of a variety of glycopeptides (see the SusD structure in Fig. 3 and supplemental figure 3) via recognition of the terminal Gal sugar and present Supplemental Fig 4 as a model to demonstrate how these glycans could be presented to BT4247 SusC transporter. The BT4246-BT4247 model, using our crystal structure and an AlphaFold 3 model of BT4247, was created based upon an overlay of each structure with PDB 6ZLT, which features the open conformation of the SusCD-like proteins BT1762 and BT1763 for the uptake of levan.

Minor:

Title: mucin-O-glycans: please italicize the O here and elsewhere in the manuscript per formal chemical nomenclature –

Corrected.

N-acetylgalactosamine: please italicize the N here and elsewhere.

Corrected.

L32, “mucin O-glycan glycoprotein”, repetitive omit glycan and write “O-glycoprotein”

Corrected.

L49, “rigid and impenetrable”: compact may be more precise. Not sure if rigidity per se has been reported, and impenetrable is only valid for bacteria not smaller metabolites! Please rephrase for clarity.

Corrected.

L69: Not sure what is meant by "must be coordinated". Perhaps balanced would be a better wording, as fiber starvation triggers increased mucin breakdown and loss of balance, so there is no coordination as such, just a balance, that can be disrupted by dietary or clinical factors. In fact these three words could be omitted altogether, since the balance is clarified in the next sentence

Corrected.

L71, consider using systemically instead of "distally"

Corrected.

L102, "seen exhibit": Lingo, seems to lack a "to" or something

Corrected.

Results:

L131, why is *Sus* in italics?

Not sure! Has been de-italicised.

L140, check lingo please, "surface that" appears wrong, swap to which

Corrected

L141: Please define "small oligomers", i.e. indicate a DP, it has been shown that such proteins target large oligomers more efficiently than smaller counterparts.

This has been corrected with new references added.

L150: Can the authors comment on why there is still some signal in the KO staining? The polyclonal antibodies are expected to target the CBM32, which is joint to many proteins from this bacterium including the target SGBP? Have the authors investigated the level of cross-reactivity of the antibody to robustly argue against the signal being at least in part stemming from other CBM32 containing proteins, which may be induced by the complex substrate? Please comment, and maybe indicate shared aa % identity.

Apologies for the confusion, but the polyclonal antibodies used in the assays were raised against the M60-like domain of BT4244 alone and not the full protein and hence they will not bind the CBM32 domain. Before the cellular localisation assays, we also confirmed the specificity of the antibodies by testing against M60-like domains from homologues of BT4244 in the genome of *B. thetaiotaomicron*, including BT3015 (which is also induced during mucin metabolism – see Table S4) and BT4272, yet there was no detectable cross reactivity (see blots below and new Suppl Fig 1B). The antibodies were therefore highly specific and the small signal

detected in the KO strain is likely very weak non-specific binding due the high sensitivity of the approach

SDS PAGE of M60-like domains from various M60-like-domain containing proteins(BT4272, BT3015) induced during mucin utilisation in B. theta

Detection of M60-like domains using polyclonal antibodies raised against BT4244 M60-like domain shows specificity of the antibodies for the BT4244 M60-like domain

L157, the Fab α and the Fc α should be labelled in the figure and their sizes indicated to make it clear that the cleavage fits the size of the domains in the cited Fig. 2F panel.

Now corrected. The IgA structure is now labelled showing various regions. For further clarity and simplicity, we have also kept the labelling to Fc and Fab only (across the entire manuscript)

L177-178, the Ka is lower on the GalNAc, so what is the basis to claim preference for the amino sugar?

We apologise but this is due to a mix up as we had the data the wrong way round in Table S1. This has now been corrected.

L183-183, please mention which structures are AlphaFold models to distinguish them from experimental crystallography models so that is clear to the reader.

Information added. All AlphaFold model structures are now indicated in the figure using asterisks.

L192. Galactose-configured may be clarified to “..with a terminal Gal”

Corrected. This has been changed through out the text

L211-213, no Fig. panel is cited for the density, so please cite the right panel.

Corrected.

L217, insert space between protein(Fig. 3F)

Corrected.

L235, the kcat appears low, would be interesting to try a galactobiose b-Gal1,3Gal, to assess of the enzymes efficiency is better than on the HexNAc?

The reviewer makes a good point and we have now tested the enzyme against beta1,3-galactobiose (a plant arabinogalactan derived sugar) to assess the preference for a non-HexNAc at +1, while retaining the b1,3 bond (unlike the lactose data shown). The data reveal that while the enzyme is active against beta1,3-galactobiose, the catalytic efficiency is ~4 fold lower (driven by both lower kcat and higher Km) than for Core 1, supporting the mucin-derived sugar as a preferred substrate for the enzyme. A comment on this is made in the revised MS in the description of the activity of the galactosidase (line 261).

L247, L257-259, Interesting with the activity on the Tn antigen. However, the release of the GalNAc is not obvious from the TLC in Fig.4D, which lacks a GalNAc standard, please clarify or even better include the GalNAc as a standard to make this clear to the reader. Is this activity comparable to the activity on the blood group A antigen? Or is this a minor trace activity? Would be splendid if enzyme concentration, incubation temperature, and time are mentioned in the legends of this and all similar figures to evaluate the levels of turnover without referring to the methods.

We agree this was not clear and we have also addressed this in response to Reviewer1's similar comment. The original image which contained the GalNAc standard and Tn antigen standards was cropped. It has now been uncropped to show the standards and this info added to the legend. Notice that released GalNAc co-migrates with Gal and is masked because the Gal stains more intensely than GalNAc. Evidence of BT4243 GalNAc'ase activity in Fig 4D is supported by the disappearance of the Tn antigen (GalNAc-Ser) substrate rather than the production of GalNAc, but also the BSM data (now Fig. 4E) which shows clear release of GalNAc from the de-sialylated mucin. The relative activity of BT4243 vs Tn and Blood group A is shown in Fig 5 where a time course of the reaction with the same amounts of enzyme and substrates are shown. The activity of BT4243 vs Tn looks somewhat lower than for BgA, but not dramatically. Info on the enzyme concentrations, temp and times have now been added to the legend as requested.

L262, Space between genus and species

Corrected

L269, fix ref!

Corrected

Figs.

Fig. 1A: Please make sur that there is key in the figure or legend to all things indicated. The different cell types, sulphate linkages, mucin domain are all not defined, please clarify this figure for the reader.

Names for various items in the figure have now been included in the figure and figure key.

Fig. 2A: Is the cartoon representation of the enzymes under the gene locus drawn in scale?. Please clarify in the legend.

Yes these are drawn to scale. This has now been clarified in the legend and the length of each protein added to the figure in brackets after the cartoon.

Fig.2B,C please label the markers with the Mw of the different bands for the Coomassie and the western staining, so the reader can appreciate the size of the band, not any obvious where to look in the Coomassie stained gel. The resolution is not good or the same in the C panel, so imperative with the labelling of the marker bands!

Markers/marker sizes have now been included

Fig. E. Neat experiment, would have been nice to see the marker and to indicated the expected size of the fragment though.

Markers/marker sizes have now been included

Fig. 3: The use of colors is a bit inconsistent in this figure. Why are the carbon atoms in black in panels D and E and blue in F and G. Please color only the carbon atoms and not all atoms in panel C, i.e. maintain the elemental color code to make the chemistry of the side chain and ligand easier to spot for the reader.

This figure has now been revised to be clearer and with more consistent labelling including maintaining the elemental colouring of the ligands and side chains. The BT4246 vs canonical SusD comparison has now been revised and moved to Suppl Fig. 3B to provide more space in the main Figure and hopefully make things clearer.

Fig. 7A: The black arrow which assigns of the cleavage site of the glycopeptidase is confusing. This protease prefers de-sialylated Tn/T antigens?. Similarly, why has the arrow of the sialidase been placed on the 2-3 sialyl? I guess this sialidase is also active on other linkages? Please correct/clarify.

The figure has now been revised to show a partially de-sialylated IgA as the preferred substrate and this info has been added to the legend. The sialidase is indeed active on both 2,3 and 2,6 linkages and we thank the reviewer for spotting this omission - this has now been shown on the figure.

REVIEWERS' COMMENTS

Reviewer #1 (Remarks to the Author):

The authors of this manuscript have done an admirable job of addressing the comments from all reviewers. I appreciate them spending time to focus on aspects I highlighted that improve the readability of the work.

In your response to my comments I think you mixed up operon 1 and operon 2, but this seems to be correct in the actual paper.

Thank you also for clarifying the TLC aspect, although I think that there is a way to optimise for staining of amino sugars. The aniline stain you use has been shown to work well to stain GlcNac for example, but different running buffers (replacing the acetic acid with ethanol) can also be used that give better resolution with amino sugars and charged sugars.

We thank the reviewer for the very positive remarks and ideas.

Reviewer #3 (Remarks to the Author):

I would like to thank the authors for the great job revising the manuscript, which is reads much better.

We thank the reviewer for their positive remarks about the revised MS and appreciate their significant contribution to the review process.

I have two issues which might be important to address in the final revision:

1) L432-434: The sentence seems to imply that SusD recognizes the terminal Gal, but is indiscriminate to the rest of the glycopeptide. As stated in my first revision, the claim is not warranted based on the crystal soaks with oligosaccharides, which are not relevant substrates. The model added in the revision (SI Fig. 4) unfortunately does not add support to the mode of binding of glycopeptides as the glycopeptide is not modelled/docked. Therefore, I strongly advise the authors to nuance this claim, which is overshadowed by the lack of explanation for the mode of selection of a preferred size range of glycopeptides for uptake by the transport system. Maybe substitute with a sentence à la “The recognition of the terminal Gal is key.., but additional interactions with the SusC or SusD may facilitate the selection of preferred uptake glycopeptides.” Or something along these lines.

We agree with the reviewer’s point and have modified the text as follows (lines 426-430):

‘While recognition of terminal Gal-containing glycopeptides by BT4246-SusD like could provide a mechanism for glycopeptide uptake that does not require binding to the variable peptide sequence, additional interactions between the peptide and the SusC or SusD may also facilitate the selection of preferred glycopeptide substrates (Supplemental Fig. 4).’

We have also added a similar caveat to the end of the legend of SI Fig 4: “While recognition of a terminal Gal configured sugar could provide a mechanism for selection of a range of glycopeptides substrates, there may also be additional interactions between the SusC and/or

SusD and the peptide that could also contribute to selection of the preferred glycopeptide substrates.”

2) L479-481: “While these findings... may play a role in UC”

No causative links are shown, but merely associations. Please tone down to what is permissive by the data, which finds no consistent or compelling support based on the following discussion.

This has now been done and now reads as below (lines 476-479):

‘While these findings are intriguing and suggest that the underrepresentation of the three *Bacteroides* species encoding the PUL BT4240-50 and its two related PULs, may have an association in some way with UC, interpretation is limited by small sample size, dataset noise, and the inability to confirm active live cell counts’

Please have a look at a few minor issues in the revised version:

L32: Please correct to mucin-O glycoproteins to “mucin O-glycoproteins”

This has now been done.

The abstract could benefit for a round of focusing and shortening.

This has now been done.

L241: Add Å to the RMSD.

This has now been done.

Reviewer #4 (Remarks to the Author):

The authors have done an excellent job incorporating the suggested experiments and addressing all concerns. The additional data and clarifications strengthen the study, making the findings even more robust. The explanation for not including NMR is also well-reasoned and clearly justified.

With these revisions, the manuscript is now clear, well-structured, and suitable for publication in Nature Communications.

We thank the reviewer for the very positive remarks about the revised manuscript and the findings presented.